# SELF-ALIGNMENT FOR OFFLINE SAFE REINFORCEMENT LEARNING

## ABSTRACT

Deploying an offline reinforcement learning (RL) agent into a downstream task is challenging and faces unpredictable transitions due to the distribution shift between a offline RL dataset and a real environment. To solve the distribution shift problem, some prior works aiming to learn a well-performing and safer agent have employed conservative or safe RL methods in the offline setting. However, the above methods require a process of retraining from scratch or fine-tuning to satisfy the desired criteria for performance and safety. In this work, we present a simple model-based RL method with a transformer and a world model, and propose a Lyapunov conditioned self-alignment method, which does not require retraining and conducts the test-time adaptation for the desired criteria. We show that our model-based RL with the transformer architecture can be described as a model-based hierarchical RL. As a result, we can combine hierarchical RL and in-context learning for self-alignment in transformers. The proposed self-alignment framework aims to make the agent safe by self-instructing with the Lyapunov condition. In experiments, we demonstrate that our self-alignment algorithm outperforms safe RL methods in continuous control and safe RL benchmark environments in terms of return, costs, and failure rate.

## 1 INTRODUCTION

Ensuring safety in real-world online reinforcement learning (RL) is crucial to making recent advances in deep RL algorithms (Haarnoja et al., 2018; Janner et al., 2019; Lee et al., 2023; Eysenbach et al., 2022) more practical, especially when a *downstream controller* (RL agent) suffers from underactuated robotics (Tedrake, 2009) or is deployed in the wild. Offline RL (Kumar et al., 2020) studies have shown that RL agents can be pretrained with well-curated offline RL datasets with human supervision, such as D4RL (Fu et al., 2021b), RL-unplugged (Gulcehre et al., 2020), and DSRL (Liu et al., 2023a) to learn better-performing and safer policy by utilizing the existing data. However, deploying a pretrained offline RL model naively without considering the many facets of unknown test-time environment is not sufficient to guarantee the safety of downstream controller. Recent two results (Ghosh et al., 2022b; 2021) have provided some insight into the challenge that a downstream controller suffer from by highly uncertain and partially observable test time environment. These studies point out that, even for the same observation, the transition probability can be unpredictable due to the uncertain nature of the system's dynamics at each step. Hence, specifying and adapting the environment transition during test-time for an RL agent is necessary to avoid risky consequence. (Ghosh et al., 2022b) learned explicit belief as an augmented input for policy to adapt the test-time environment can help an downstream controller to be safer.

Belief based adaptation, however, requires its own pretraining algorithm from scratch for test-time adaptation. It is hard to align the pretrained agent to be safe without additional fine-tuning procedure. Self-alignment by leveraging the pretrained distribution from offline RL dataset could be an easier way to deploy an agent more safely. Self-alignment (Sun et al., 2023) is one of the alignment approaches for Large Language Models (LLMs), which induces desirable outputs for specific instruction prompts. It enables efficient adaptation of LLMs for a particular purpose without fine-tuning by utilizing the reasoning and generative power of transformer-based large models. To apply self-alignment to model-based RL agent, we use the proposed transformer-based architecture which is composed of an agent and a world model to learn policy and predictive model simultaneously, by generating a virtual imagination of agent trajectory.

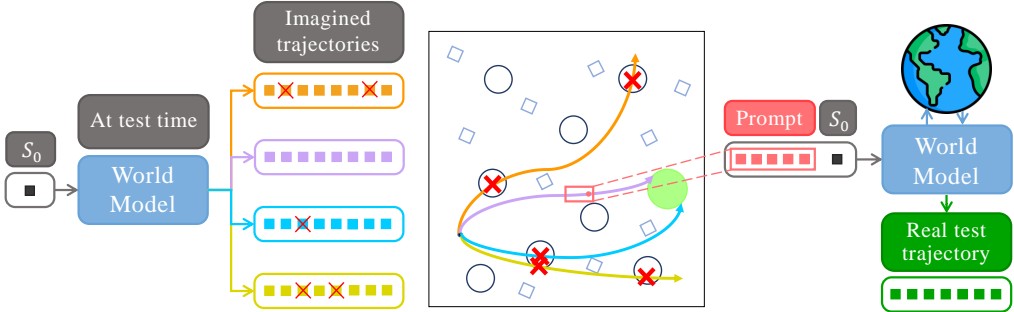

Figure 1: An illustration of the three stages of our Self-Alignment for offline safe RL. We illustrate one of the safety gym environments **in the middle box,** where small circles and squares represent hazards, and a green circle indicates the goal region. At test time for the downstream task, we first use the RL transformer to generate several imagined trajectories for a given initial state. Secondly, we compute the occupancy measure and the Lyapunov condition of state-action pairs in imagined trajectories to determine which trajectory violates the Lyapunov condition the least. For example, ✖ denotes a $(s, a)$ pair that incurs a high cost and violates the Lyapunov condition simultaneously. Finally, we retrieve the best trajectory segment from the candidate imaginations for prompt and augment the retrieved segment and initial state for self-instruction at test time.

Transformer-based RL has shown the ability of prompt-based alignment, which enforces transformers to conduct in-context learning and produce a desirable behavior for a given prompt. For example, training multi-modal prompts, which consist of text, video, and trajectory data (Jiang et al., 2023a), enables the model to solve various robot manipulation tasks and shows remarkable generalization capability for unseen complex tasks. However, the specific structure of the training input data, where the prompt and trajectory tokens lie in consecutive order, is needed to adapt to a newly defined task.

In this work, we propose a *self-alignment technique by self-generated prompt* to guarantee the better safety. Our self-generated prompt for safety is based on *Lyapunov condition*. To implement self-alignment for safety, we present a novel formulation of Lyapunov condition as a probabilistic inference and transformer-based RL world model as a *model-based hierarchical RL agent*, respectively, to provide in-context learning based self-alignment. We present an overview of our algorithm, which we call self-alignment for safety (SAS), in fig. 1. First, the proposed transformer-based model with the agent and the world model generates several imagined trajectories using the learned policy and predictive model from the data distribution. We evaluate the safety using *the proposed inference model of the Lyapunov condition*, and feed the most likely trajectory in terms of the Lyapunov stability condition into a prompt of our model. The given prompt instructs our model to act in accordance with its Lyapunov condition property. We explain this ability of our transformer-based architecture as a skill-conditioned hierarchical RL in section 5.1. In our experiments, we demonstrate the efficiency and safe deployment of SAS in 12 Safety Gymnasium environments (Ji et al., 2023) and OpenAI Gym Mujoco (Brockman et al., 2016). SAS outperforms prior safe RL methods by up to 2 times on Safety Gymnasium benchmarks and 2 times on Mujoco in terms of failure rate.

## 2 RELATED WORK

**Transformer-based RL.** Transformer-based RL (Janner et al., 2021; Chen et al., 2021) has been emerged by making a connection between pretraining of GPT (Radford et al., 2018) and offline RL with prior data. Recently, several model-based RL methods with transformers, which are called *world model*, lead to sample efficient online RL by leveraging the structure of auto-regressive generation in terms of imagination in model-based RL, such as TWM (Robine et al., 2023) and IRIS (Micheli et al., 2023). For predictive model, TWM applies VAE (Kingma & Welling, 2013), and IRIS uses VQGAN (Esser et al., 2021) to reconstruct the observation. Prompting on transformer-based RL was also proposed to help task specification with multi-model prompts (Jiang et al., 2023b), and achieve test-time adaptation by learning with prompts from scratch (Xu et al., 2022b). CDT(Liu et al., 2023b) is similar to our work since CDT uses decision transformer for offline safe RL by modifying the DT architecture to feed the cost value to train an offline safe RL agent. Unlike prior works, we aim

to align *the transformer for model-based RL* by providing self-generated instruction for in-context learning without any fine-tuning.

**Safe RL and Lyapunov condition.** Lyapunov condition has been applied to safe control (Chang et al., 2019) and safe RL (Chow et al., 2018) in many different ways. LDM (Kang et al., 2022) proposed an integration of Lyapunov condition and offline RL to avoid distribution shift for safety. While safe RL algorithms are usually formulated as constrained MDP, which introduces a control barrier functions to prevent an RL agent from entering unsafe regions (Bansal & Tomlin, 2021; Ganai et al., 2023; Kim et al., 2023), we instead focus on validating Lyapunov condition for safety to avoid unsafe regions caused by the distribution shift (Tedrake, 2009; Bharadhwaj et al., 2020). DCRL (Qin et al., 2021) is similar to ours in online safe RL, which employs a constraint on the level of state density to stay in the highly probable states. In contrast, SAS does not require a constrained RL tuning for transformer or cost, and adapt a safe RL task by self-alignment at test time.

**Large model alignment.** Alignments in LLMs have been proposed to learn human preference or make pretrained models safer and more helpful recently (Ouyang et al., 2022). For example, a pretrained general language assistant can be aligned to be *helpful, honest, harmless* (HHH) (Askell et al., 2021). Alignment methods for LLMs can be classified into RLHF (John Schulman, 2022) and instruction based in-context learning (Sun et al., 2023; Wang et al., 2023). Alignment by instruction is an emerging technique to align large language models (LLMs) output with a specific desired behavior by engineering instruction prompt (Brown et al., 2020), RLHF (Ouyang et al., 2022), and zero-shot reasoner (Kojima et al., 2022). In RL, aligning Large Models (LMs) from pretrained distribution is natural and well-behaved in LLMs for human preference, but very limited for unseen task specification by demonstration (Jiang et al., 2023a) and learning for augmented prompt (Xu et al., 2022b), even though alignment for safety is essential to ensure safety in real-world RL.

## 3 PRELIMINARIES

**Problem Setting.** We consider a discounted Markov Decision Process $\langle \mathcal{S}, \mathcal{A}, \mathcal{R}, \mathcal{C}, \mathcal{P}, \mathcal{P}_{\mathcal{S}_1}, \gamma \rangle$, where $\mathcal{S}$, and $\mathcal{A}$ are observation and action spaces, $\mathcal{R} : \mathcal{S} \times \mathcal{A} \times \mathcal{S} \to \mathbb{R}$ and $\mathcal{C} : \mathcal{S} \times \mathcal{A} \times \mathcal{S} \to \mathbb{R}$ are reward and cost functions, $\mathcal{P} : \mathcal{S} \times \mathcal{A} \to \mathcal{S}$ is the transition operator, $\mathcal{P}_{\mathcal{S}_1} : \mathcal{S} \to [0, 1]$ is the initial state distribution, and $\gamma \in (0, 1)$ is the discount factor. We first define latent skill space $\mathcal{Z}$ and consider two following hierarchical policies. The high-level policy $\pi_\theta^{\text{high}} : \mathcal{S} \times \mathcal{Z} \to \mathcal{Z}$ with parameter $\theta$ for skill selection maps the previous latent skill and the observation to the choice of learned skills $\mathbf{z} \in \mathcal{Z}$ from the pre-collected experience. The low-level policy $\pi_\phi^{\text{low}} : \mathcal{S} \times \mathcal{Z} \to \mathcal{A}$ with parameter $\phi$ interacts with environment with given skill in the agent's action space.

**Density Constrained Safe RL.** From a conservatism, constraining the density of states and actions has been studied to enhance the agent safety. Lyapunov Density Model (LDM) (Kang et al., 2022) and DCRL (Qin et al., 2021) are offline and online safe RL frameworks, which design the constraint regions based on density. In the case of offline safe RL, LDM leverages Lyapunov stability condition to constrain the density of state-action pairs over a *long horizon*. Finding a control Lyapunov function for arbitrary MDP is a challenging problem. To tackle this issue, LDM provides a modified version of Bellman operator which can be interpreted as learning a control Lyapunov function by the offline data. Specifically, this learning process stitches policies toward more probable terminal states. To guarantee that an offline RL agent does not escape from sinking into the low density region, we first introduce the control invariant set and the condition of Lyapunov model of LDM as follows:

**Definition 3.1.** Let $(\mathbf{s}_e, \mathbf{a}_e)$ be an equilibrium point and $\tau = ((\mathbf{s}_1, \mathbf{a}_1), \cdots, (\mathbf{s}_e, \mathbf{a}_e))$ be an Lyapunov stable trajectory. For all Lyapunov stable trajectories $\tau$, LDM $G(\mathbf{s}_t, \mathbf{a}_t)$ must satisfy the following:

**(1)** $G(\mathbf{s}_e, \mathbf{a}_e) = 0$, **(2)** $G(\mathbf{s}_t, \mathbf{a}_t) > 0, \forall (\mathbf{s}_t, \mathbf{a}_t) \neq (\mathbf{s}_e, \mathbf{a}_e)$, **(3)** $G(\mathbf{s}_t, \mathbf{a}_t) \geq G(\mathbf{s}_{t+1}, \mathbf{a}_{t+1})$.

To learn a valid control Lyapunov function, the LDM backup operator is defined as

$$\mathcal{T}_{\text{LDM}} G(s, a) = \max\{-\log \rho(s, a), \gamma \min_{a'} G(\mathcal{P}(s, a), a')\} \tag{1}$$

where $\rho(s, a)$ is a density of given state and action. For a Lyapunov density model $G(\mathbf{s}_t, \mathbf{a}_t)$, there exists a control invariant set for a constant $c > 0$:

$$\mathcal{R}_c = \{(\mathbf{s}_t, \mathbf{a}_t) | G(\mathbf{s}_t, \mathbf{a}_t) \leq c\}.$$

Intuitively, a greedy policy of the Lyapunov density model, $\arg\min_a G(s,a)$ does not escape from the control invariant set $\mathcal{R}_c$ and eventually reaches the equilibrium point where, by the definition, has the highest density value in the control invariant set $\mathcal{R}_c$.

# 4 MODELING OFFLINE SAFE RL AS DENSITY-BASED LYAPUNOV CONTROL

In this work, we introduce a transformer network based on the *Decision Transformer* (Chen et al., 2021) architecture and extend this to include a VAE (Kingma & Welling, 2013) based predictive model to incorporate imaginations for searching safe candidate trajectories within the model-based RL framework. The detailed architecture of our transformer model is described in table 5. Since we pretrain our transformer using the offline RL dataset $\mathcal{D}$, we utilize learned predictive model distribution and the policy distribution to compute an approximate Control Lyapunov function which is analogous to LDM. Rather than using one-step density value, we use the occupancy measure estimate $\hat{\rho}$ by computing the following equation:

$$\hat{\rho}(s,a) = \sum_{t=0}^{\infty} \gamma^t \rho(\mathbf{s}_t = s, \mathbf{a}_t = a | P_{\mathcal{S}_1}, \pi, \mathcal{T}) = \sum_{t=0}^{\infty} \gamma^t \rho_{\text{VAE}}(\mathbf{s}_t = s | P_{\mathcal{S}_1}, \pi, \mathcal{T}) \pi(\mathbf{a}_t = a | \mathbf{s}_t = s),$$

where $\rho_{\text{VAE}}$ and $\pi$ denotes the density estimation of VAE and the policy. We note that the occupancy measure estimate $\hat{\rho}$ can be computed from the generated trajectory $\tau$ of autoregressive transformer.

To capture the connection between the density and offline safe RL, we introduce safe RL problem as the constrained optimization problem as follows:

$$\max_{\pi} \ J_R(\pi) \quad \text{s.t.} \ J_C(\pi) \leq d, \tag{2}$$

where $J_R(\pi), J_C(\pi)$ are the expected discounted sum of reward/cost functions, respectively. We reformulate eq. (2) in terms of occupancy measure as

$$\max_{\pi} \mathbb{E}_{s,a} \left[ \rho^\pi(s,a) r(s,a) \right] \quad \text{s.t.} \quad \mathbb{E}_{s,a} \left[ \rho^\pi(s,a) C(s,a) \right] \leq d.$$

Let $U$ be the universal set of state-action space and $B$ be the set $\{(s,a) | (s,a), C(s,a) < C_{th}\}$ where $C_{th}$ is the some threshold of cost which satisfies $C_{th} \leq d(1-\gamma)$. Assume that the cost value is bounded as $0 \leq C(s,a) \leq C_{max}$. We define a volume constant $\alpha = \mathbb{E}_{(s,a)\sim U}[\mathbf{1}((s,a) \in B)]$, which is $0 < \alpha < 1$. Then, we can write the above inequality as

$$J_C(\pi) = \mathbb{E}_{(s,a)\sim B} \left[ \hat{\rho}^\pi(s,a) C(s,a) \right] + \mathbb{E}_{(s,a)\sim B^C} \left[ \hat{\rho}^\pi(s,a) C(s,a) \right]$$
$$J_C(\pi) - d = \mathbb{E}_{(s,a)\sim B} \left[ \hat{\rho}^\pi(s,a) C(s,a) \right] - \alpha d + \mathbb{E}_{(s,a)\sim B^C} \left[ \hat{\rho}^\pi(s,a) C(s,a) \right] - (1-\alpha)d$$
$$\leq \mathbb{E}_{(s,a)\sim B} \left[ \hat{\rho}^\pi(s,a) C_{th} - d \right] + \mathbb{E}_{(s,a)\sim B^C} \left[ \hat{\rho}^\pi(s,a) C_{max} - d \right]. \tag{3}$$

We can observe that expert policies in the offline RL dataset $\mathcal{D}$ should have low values in $B^C$ to satisfy the constraint inequality of eq. (3) less than zero. Now, we know that reducing the marginal value of occupancy measure over $B^C$ leads to the lower bound of the cost function of $\pi$. Suppose that $\hat{\rho}^\pi(s,a) \leq \frac{d}{C_{max}}$ for $(s,a) \in B^C$. We consider the definition of occupancy measure and assume that occupancy measure is a continuous function. We have $\frac{d}{C_{max}} \leq \hat{\rho}^\pi(s,a) \leq \frac{d}{C_{th}} \leq \frac{1}{1-\gamma}$ for $(s,a) \in B$, and then get $J_C(\pi) - d \leq 0$. This is an intuitive condition to be an expert policy trained by the constrained RL in eq. (2). Now, we generalize the above condition into more general offline RL scenario. We consider the occupancy measure of the given offline data, $\rho_{\text{data}}(s,a)$ and the occupancy measure of optimal policy $\hat{\rho}^{\pi*}$ satisfies the single policy concentrability:

$$\hat{\rho}^{\pi*}(s,a)/\rho_{\text{data}}(s,a) \leq D$$

where all $(s,a) \in S \times A$, and $D = \max_{\pi} D^\pi$ represents the widely-used uniform concentrability coefficient. This assumption, drawn from (Rashidinejad et al., 2021), is used to incorporate various sources of offline RL data, including medium-level datasets. To prevent having overestimated density in the region which might lead to failure, we insert the concentrability coefficient as a margin for defining the target control invariant set under the pretrained distribution,

$$R_\rho = \{(s,a) | \rho_{\text{data}}(s,a) \geq \frac{d}{C_{max} D}\}. \tag{4}$$

This implies that the offline RL agent can avoid moving toward $(s, a) \in B^C$ by applying a penalty using $D$. Our key idea is to search a density-based Lyapunov stable trajectory sample by the imagination process of our model-based RL transformer. We first define the Energy $E = -\log \hat{\rho}_{\text{data}}$ for convenience. From eq. (1), we define our approximate Lyapunov model as

$$G_{\text{SAS}}(\mathbf{s}_t, \mathbf{a}_t) = \log \hat{\rho}_{\text{data}}(\mathbf{s}_t, \mathbf{a}_t) - \max_{\pi} \min_{t'} \log \hat{\rho}_{\text{data}}(\mathbf{s}_{t'}, \pi(\mathbf{s}_{t'})) \tag{5}$$

$$= \min_{\pi} \max_{t'} E(\mathbf{s}_{t'}, \pi(\mathbf{s}_{t'})) - E(\mathbf{s}_t, \mathbf{a}_t). \tag{6}$$

where $\hat{\rho}_{\text{data}}$ is the learned distribution of the offline RL dataset distribution $\rho_{\text{data}}$. By randomly sampling using the VAE and the stochastic policy of our transformer, we generate multiple trajectories from imagination and compute the optimal sample based on eq. (5) across trajectories and time. Note that repeating to get more samples induces the tighter upper bound of the control invariant set having more probable actions. Finally, we define the target control invariant set in terms of $\rho_{\text{data}}(s, a)$ as

$$\mathcal{R}_G^{\text{SAS}} = \{(\mathbf{s}_t, \mathbf{a}_t) | G_{\text{SAS}}(\mathbf{s}_t, \mathbf{a}_t) \leq -\log \frac{d}{C_{\max} D}\}. \tag{7}$$

We note that the cost condition of $d$ is applied to the density constraint in terms of control invariant set. Now, we solve a Lyapunov stable policy in terms of $G_{\text{SAS}}$, then have the offline safe RL policy.

## 4.1 DENSITY-BASED LYAPUNOV CONTROL AS PROBABILISTIC INFERENCE

To search and infer safe imagined trajectory samples, we propose the probabilistic inference formulation of Lyapunov condition in Definition 3.1.

**Theorem 4.1** (Lyapunov Condition Observable). *Let two observables $\mathcal{U}_t$ and $\mathcal{V}_t$ be the indicator variables*

$$\mathcal{U}_t = \mathbf{1}\left[G_{\text{SAS}}(\mathbf{s}_t, \mathbf{a}_t) > 0\right], \quad \mathcal{V}_t = \mathbf{1}\left[G_{\text{SAS}}(\mathbf{s}_t, \mathbf{a}_t) - G_{\text{SAS}}(\mathbf{s}_{t+1}, \mathbf{a}_{t+1}) \geq 0\right]. \tag{8}$$

*The problem of finding a trajectory from Lyapunov stable controller is equivalent to solve the following inference problem:*

$$\max_{\tau} \frac{1}{T} \sum_t \log \left(P(\mathcal{U}_t = 1|\tau)P(\mathcal{V}_t = 1|\mathcal{U}_t = 1, \tau)\right), \tag{9}$$

*where $T$ is the length of trajectory $\tau$.*

See appendix E for proof. We note that the first Lyapunov condition observable $\mathcal{U}_t$ is indirectly computed by lines 1 to 6 in Algorithm 1. In the first loop of Algorithm 1, among the $N$ iterations, we select the episode with the lowest maximum energy value reached by each imagined trajectory. In line 6, we set the selected lowest maximum energy $\hat{E}_j$ as the value of our approximate Lyapunov model for the equilibrium point, $G_{\text{SAS}}(s_e, a_e) = \min_{\pi} \max_{t'} E(s_e, a_e) - e(s_e, a_e) = \hat{E}_j - E(s_e, a_e) = 0$. Then, all other $N - 1$ episodes have steps with $G(s, a) < 0$ inevitably due to $G_{\text{SAS}}(s_e, a_e)$, leading to violation of the condition $U_t = 1$. To search for a Lyapunov stable policy which guarantees all state-action pair elements are in $R_G^{\text{SAS}}$ in Equation 7, we assume that a test-time agent can access a set of previously learned policies, $\Pi = \{\pi_i\}_{i=1}^N$ from pretrained distribution. Each generated trajectory at $i$-th iteration, $\tau_i$, corresponds one-to-one to a certain $\pi_i \in \Pi$ at given initial state $s_0$. As we increase the number of iterations of the first loop, $N \to \infty$, we can get a lower $\hat{E}_j$, which leads to the more probable subsequent $(s, a)$ and equilibrium point. Furthermore, selecting the most probable index $k^*$ in the second for-loop implies that we choose the optimal-selection policy which violates the condition $\mathcal{V}_t$ least. Then, the selected $\pi$ is highly likely to satisfying in the control invariant set $\{(s, a)|0 \leq G_{\text{SAS}}(s, a) \leq \hat{E}_j\}$. Now, we rewrite Equation 6 for Algorithm 1 as

$$G_{\text{SAS}}(\mathbf{s}_t, \mathbf{a}_t) = \log \hat{\rho}_{\text{data}}(\mathbf{s}_t, \mathbf{a}_t) - \max_{i=1,\cdots,N} \min_{j=1,\cdots T} \log \hat{\rho}_{\text{data}}(\mathbf{s}_j, \pi_i(\mathbf{s}_j))$$

$$= \min_{i=1,\cdots,N} \max_{j=1,\cdots T} E(\mathbf{s}_j, \pi_i(\mathbf{s}_j)) - E(\mathbf{s}_t, \mathbf{a}_t).$$

We now demonstrate that Algorithm 1 reduces the probability of escaping from the control invariant set as the numbers of iterations, $N$ and $M$ for the first and second loops, respectively, increase.

---

**Algorithm 1** **S**elf **A**lignment To be **S**afe (SAS). Self-generating prompt for instruction.

---

**Require:** Pretrain transformer from $\mathcal{D}$ and environment initial state $\mathbf{s}_1^{\text{test}} \sim \mathcal{P}_{\mathcal{S}_1}$

  1: **for** $i = 1, 2, 3, \ldots, N$ **do**                          ▷ for condition $\mathcal{U}_t$
  2:      Sample $\tau_i \sim p(\tau | \mathbf{s}_1^{\text{test}})$ by imagination
  3:      Compute $E_t$ of each time-step $t$
  4:      $\hat{E}_i, t_i \leftarrow \max_t E_t, \; \arg\max_t E_t$
  5: **end for**
  6: $j \leftarrow \arg\min_i \hat{E}_i$
  7: Compute an initial prompt $\tilde{\mathbf{p}}_{1:L} \leftarrow \tau_j[t_j - L : t_j]$
  8: **for** $k = 1, 2, 3, \ldots, M$ **do**                        ▷ for condition $\mathcal{V}_t$
  9:      Sample $\tau_k' \sim p(\tau | \tilde{\mathbf{p}}_{1:L}, \mathbf{s}_1^{\text{test}})$
10:      Compute $E_t$ of each time-step $t$
11:      $\hat{E}_k, t_k \leftarrow \max_t E_t, \; \arg\max_t E_t$
12:      $v_k \leftarrow \sum_t \mathcal{V}_t$
13: **end for**
14: $k^* \leftarrow \arg\max_k v_k$
15: Prompt $\mathbf{p}_{1:L} \leftarrow \tau_{k^*}'[t_{k^*} - L : t_{k^*}]$                   ▷ self-alignment

---

**Proposition 4.2** (Probability of out-of-distribution trajectory). *Assume that the sampled state action pairs $(s_t, a_t)$ in the trajectory is i.i.d. Let $\tau = \{(s_t, a_t)\}_{t=1}^T$ denote the set of state-action pairs in the trajectory with length $T$ and $\mathcal{D}$ is the pretrained distribution of expert trajectories. By Assumption E.1, the probability that the best trajectory escapes from the target control invariant set in Algorithm 1 is bounded and decreases as the numbers of iterations $N, M \to \infty$ as follows:*

$$P\left[\tau_{best} \not\subset \mathcal{R}\right] \leq \left[\frac{\mathbb{E}_{(s,a)\sim\mathcal{D}}[-\log \hat{\rho}_{\text{data}}(s,a)]}{c_2}\right]^{NT} + \exp\left(-\frac{2M\kappa^2(c_2 - c_1)^2}{TL^2}\right) \quad (10)$$

The proof is in appendix E.3.

## 5   Self-Alignment for safe RL with Lyapunov condition

When we aim to make a transformer-based model aligned for a downstream task, the model can learn the given task by conditioning a prompt which is composed of demonstration examples. This remarkable ability is called in-context learning, which can be explained by the implicit Bayesian conditional inference with demonstration prompt in NLP domain (Xie et al., 2021). The given demonstration prompt into transformer predicts an aligned output which is conditioned on the prompt. The inference probability is defined as the following posterior predictive distribution:

$$p(\texttt{output}|\texttt{prompt}) = \int p(\texttt{output}|\texttt{prompt}, \theta)p(\theta|\texttt{prompt})d\theta,$$

where $\theta$ is called *latent concept*. The latent concept $\theta$ serves as a parameter determining the transition of the hidden Markov model $p(\texttt{output}|\texttt{prompt}, \theta)$, which corresponds to a learned conditional distribution of pretraining sequence dataset on a latent concept $\theta$. Then, the conditional inference of latent concept $\theta$ on $\texttt{prompt}$ selects the parameter of $p_\theta(\texttt{output}|\texttt{prompt}) = p(\texttt{output}|\texttt{prompt}, \theta)$ and makes it possible to generate an aligned output. In this section, we formulate our transformer as a hierarchical RL and decompose the policy into two policies, high-level and low-level policies. Analogous to the above implicit Bayesian concept, we now assume that the parameter of high-level policy corresponds to the latent concept. All proofs in appendix E.

### 5.1   Model-based RL with transformer as Probabilistic Inference

We extend the probabilistic graphical model of skill-based hierarchical RL to describe our model. We show that the pretrained transformer can implicitly perform Bayesian inference. To define the world

model as a HMM, we consider a probability distribution of trajectory $\tau$ as

$$p(\tau) = p(\mathbf{s}_1) \prod_{t=1} p(\mathbf{s}_{t+1}|\mathbf{s}_t, \mathbf{a}_t)p(\mathbf{a}_t|\mathbf{s}_t, \mathbf{z}_t)p(\mathbf{z}_t|\mathbf{s}_t, \mathbf{z}_{t-1}).$$

where $p(\mathbf{s}_{t+1}|\mathbf{s}_t, \mathbf{a}_t)$ denotes the transition probability of transformer, and we abuse the notation $p(\mathbf{z}_1|\mathbf{s}_1, \mathbf{z}_0)$ as $p(\mathbf{z}_1|\mathbf{s}_1)$ for brevity and clear notation. To understand this graphical model as a hierarchical RL, we consider the hidden layer of transformer as a *latent skill variable* $\mathbf{z}_t$. To show that the transformer has the property of in-context predictor, we also define the conditional probability $p(\tau|\mathbf{p}_{1:L}, \mathbf{s}_1^{\text{test}})$ where $\mathbf{s}_1^{\text{test}}$ is the initial state at test-time, $\mathbf{p}_{1:L}$ is a prompt demonstration with length $L$, as

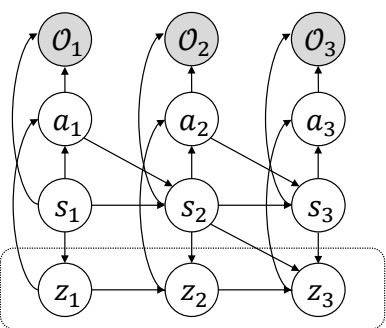

$$p(\tau|\mathbf{p}_{1:L}, \mathbf{s}_1^{\text{test}}) = \int_\theta p(\tau|\mathbf{p}_{1:L}, \mathbf{s}_1^{\text{test}}, \theta)p(\theta)d\theta, \quad (11)$$

where we define $\theta$ as the parameter of high-level policy $\pi_\theta^{\text{high}}$. At test time, we generate an output trajectory starting from $\mathbf{s}_1^{\text{test}}$ by predicting the first latent skill variable $\mathbf{z}_1^{\text{test}}$ with the prompt demonstration $\mathbf{p}_{1:L}$ and $\theta$. We can write the conditional probability $p(\tau|\mathbf{p}_{1:L}, \mathbf{s}_1^{\text{test}}, \theta)$ for a given $\theta$ as

Figure 2: The probabilistic graphical model of model-based hierarchical RL. Our model is a HMM where $\mathcal{O}_t$ is the optimality variable that correspond to $P(\mathcal{U}_t = 1, \mathcal{V}_t = 1)$.

$$\tau \sim \sum_{\mathbf{z}_1^{\text{test}} \in \mathcal{Z}} \left[ p(\tau|\mathbf{s}_1^{\text{test}}, \mathbf{z}_1^{\text{test}}, \theta)p(\mathbf{z}_1^{\text{test}}|\mathbf{p}_{1:L}, \mathbf{s}_1^{\text{test}}, \theta)) \right]$$

$$= \sum_{\mathbf{z}_1^{\text{test}} \in \mathcal{Z}} \prod_{t=1} \underbrace{p(\mathbf{s}_{t+1}^{\text{test}}|\mathbf{s}_t^{\text{test}}, \mathbf{a}_t^{\text{test}})}_{\text{VAE decoder}} \underbrace{p(\mathbf{a}_t^{\text{test}}|\mathbf{s}_t^{\text{test}}, \mathbf{z}_t^{\text{test}})}_{\pi_\phi^{\text{low}}} \underbrace{p_\theta(\mathbf{z}_t^{\text{test}}|\mathbf{s}_t^{\text{test}}, \mathbf{z}_{t-1}^{\text{test}})}_{\pi_\theta^{\text{high}}} =: \sum_{\mathbf{z}_1^{\text{test}} \in \mathcal{Z}} g_{\pi_\theta}(\tau, \mathbf{z}_1^{\text{test}}), \quad (12)$$

where we abuse the notation $p_\theta(\mathbf{z}_1^{\text{test}}|\mathbf{s}_1^{\text{test}}, \mathbf{z}_0^{\text{test}}) = p_\theta(\mathbf{z}_1^{\text{test}}|\mathbf{p}_{1:L}, \mathbf{s}_1^{\text{test}})$ for clarity. It implies that we have the random skill variable $\mathbf{z}_1^{\text{test}}$ which is sampled by the given the high-level policy parameter $\theta$. We also note that $\mathbf{p}_{1:L}$ is a demonstration state-action sequence, $(\mathbf{s}_{-L+1}, \mathbf{a}_{-L+1}, \mathbf{s}_{-L+2}, \mathbf{a}_{-L+2}, \cdots, \mathbf{s}_0, \mathbf{a}_0)$. Then, the prompt can be viewed as a concatenation in front of the following trajectory in Figure 2.

Our goal is to find the safe policy $\pi_{\theta*}^{\text{high}}$ analogous to the demonstration prompt $\mathbf{p}_{1:L}$. We note that the pretrained transformer marginalize over the family of high-level policies in the offline RL dataset as in eq. (11). More specifically, the dataset $\mathcal{D}$ is composed of the trajectories from behavior polices, and then it implies that the transformer learns the distribution from the feasible high-level policy parameter space. To retrieve $\theta^*$ of the safe high-level policy $\pi_{\theta*}^{\text{high}}$ corresponding to a given prompt $\mathbf{p}_{1:L}$, we first define the optimality variable $\mathcal{O}_t$ in Figure 2 as $\mathcal{O}_t = \mathbf{1}\left[(\mathbf{s}_t, \mathbf{a}_t) \in C_t\right]$ where $C_t = \{(\mathbf{s}_t, \mathbf{a}_t)|(\mathbf{s}_t, \mathbf{a}_t) \sim \sum_{\mathbf{z}_t, \mathbf{z}_{t-1}} \pi_\phi^{\text{low}}(\mathbf{a}_t|\mathbf{s}_t, \mathbf{z}_t)\pi_{\theta*}^{\text{high}}(\mathbf{z}_t|\mathbf{s}_{t-1}, \mathbf{z}_{t-1})\}$, the set of all possible state-action pairs with $\theta^*$. We can describe the inference $p(\mathcal{O}_{\text{traj}}|\mathbf{p}_{1:L}, \mathbf{s}_1^{\text{test}})$ as follows:

$$p(\mathcal{O}_{\text{traj}}|\mathbf{p}_{1:L}, \mathbf{s}_1^{\text{test}}) = \int_\theta \sum_{\mathbf{z}_1^{\text{test}} \in \mathcal{Z}} \left( g_{\pi_\theta}(\tau, \mathbf{z}_1^{\text{test}}) \prod_{t=1} p(\mathcal{O}_t|\mathbf{s}_t^{\text{test}}, \mathbf{a}_t^{\text{test}}) \right) e^{L \cdot r_L(\theta)} p(\theta)d\theta, \quad (13)$$

where $r_L(\theta) = \frac{1}{L}\log\frac{p(\mathcal{O}_{1:L}, \mathbf{s}_1^{\text{test}}|\theta)}{p(\mathcal{O}_{1:L}, \mathbf{s}_1^{\text{test}}|\theta^*)}$. The prompt $(\mathbf{s}_{L-1}, \mathbf{a}_{L-1}, \cdots, \mathbf{s}_0, \mathbf{a}_0)$ is originally from the high-level policy with $\theta^*$. We have $\mathcal{O}_{1:L} = 1$ when $\theta = \theta^*$ is selected and get $e^{L \cdot r_L(\theta)} \to 1$. Then, we can retrieve the safe high-level policy parameter of $\pi_{\theta*}^{\text{high}}$ to the demonstration prompt $\mathbf{p}_{1:L}$ by the above selection property in eq. (13) and regenerate and execute at the test time under $\theta^*$. This differs from the original implicit Bayesian inference (Xie et al., 2021) in two ways: (1) we introduce the low-level policy $\pi_\phi^{\text{low}}(\mathbf{a}_t|\mathbf{s}_t, \mathbf{z}_t)$ term that enable the implicit Bayesian inference method to work on the RL domain with *action space*; and (2) the transformer inherits the *predictive transition model* $p(\mathbf{s}_{t+1}|\mathbf{s}_t, \mathbf{a}_t)$ to generate an imaginary trajectory coincided with the real environment.

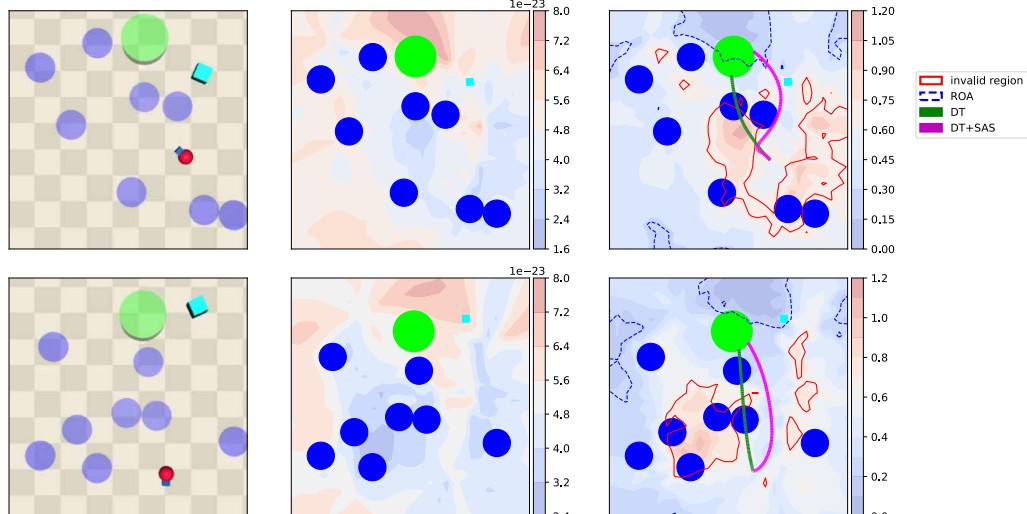

Figure 3: **SAS dodges hazard better.** We visualize `PointGoal1-v0`. **Left:** The illustrated env. has 8 fixed **hazards**, one movable obstacle **vase**, and one **goal** position. **Middle:** We visualize $\hat{\rho}_{data}(\mathbf{s})$ at each point by using our transformer. **Right:** We illustrate two trajectories without self-alignment and our DT+SAS. The landscape visualizes $G_{\text{SAS}}$ where the blue color indicates the sub-level of $G_{\text{SAS}}$. We mark Region Of Attraction (ROA) with blue dot lines, which means a forward invariant set where we can guarantee the upper bound of density. Red lines means the invalid region where exceed the 95 percentile of $G_{\text{SAS}}(\mathbf{s}_t, \mathbf{a}_t)$, which indicates unsafe region.

## 5.2 INSTRUCTION PROMPT GENERATION FOR SAFETY

Offering good exemplar in-context demonstrations (prompt) for alignment usually relies on extensive human supervision. Inspired by `Dromedary` (Sun et al., 2023) for LLMs, we align our transformer to act more stable and safer by itself without any human instruction or seed prompts. In algorithm 1, our **S**elf-**A**ligning RL agent behavior to be **S**afe (**SAS**) method involves the following procedures. **1) Lyapunov-conditioned instruction generation** provides the selection rule for Lyapunov condition to create an exemplar demonstration for reasoning a safer high-level policy $\pi_\theta^{\text{high}}$ by imagination of transformer. To generate instruction demonstration for in-context learning, we follow eq. (9) to satisfy Lyapunov condition from line 1 to 14. **2) Internal thoughts** is the generated behavior trajectory which already satisfies Lyapunov condition enough in line 7 and 14. We do not need to prepare a few in-context learning demonstration to generate internal thoughts for the final instruction. **3) Guiding the final behavior of RL transformer** is the final stage with the internal thoughts for in-context learning demonstrations to align with a safer $\pi_\theta^{\text{high}}$ by annotating with initial state in line 15.

## 6 EXPERIMENTS

We demonstrate the performance of SAS in `mujoco` (Brockman et al., 2016) and `Safety Gymnasium` (Ji et al., 2023) to evaluate the three metrics, reward return, cost return, and failure rate. We use D4RL dataset (Fu et al., 2021a) for mujoco and DSRL (Liu et al., 2023a) for safety gymnasium. We use normalization of both reward and cost returns. We denote DT as DT+SAS and CDT as CDT+SAS when we apply SAS. We modify DT (Chen et al., 2021) to predicts next state and next return-to-go as well as action. In all results, we abbreviate the task name as follows: (PointGoal1, PG1), (PointPush1, PP1) and (CarButton2, CB2). The detailed experiment setting is in appendix B.

**Does the proper internal thought make a safer decision?**  Overall, DT+SAS shows the lower cost and failure rate than DT in most environments in table 1. We note that reward may decrease as a trade-off by Lyapunov condition to reduce the aspect of pursuing high reward in DT. However, in some tasks, such as PG2, it is surprising that the reward of DT+SAS is higher than DT. In tasks, like PB1, all metrics, reward, cost and failure rate increase simultaneously. It implies that DT has trained

Table 1: Ablation study in the Safety Gymnasium. DT+rand involves inserting a random trajectory into the prompt, and DT+maxmax includes the trajectory with the argmax of the maximum value of $E$ as the prompt. **Bold**: the smallest cost among the four models. **Blue**: DT+SAS has a lower failure rate than DT. **Red**: DT+SAS has a higher cost than DT but the reward is also higher.

| Environment | | PG1 | PG2 | PP1 | PP2 | PB1 | PB2 | CG1 | CG2 | CP1 | CP2 | CB1 | CB2 |
|---|---|---|---|---|---|---|---|---|---|---|---|---|---|
| | reward | 0.660 | 0.377 | 0.218 | 0.202 | 0.379 | 0.495 | 0.638 | 0.513 | 0.35 | 0.204 | 0.237 | 0.212 |
| DT | cost | 1.319 | 2.625 | 0.927 | 0.782 | **1.188** | 1.309 | 0.976 | 1.466 | 0.678 | 1.174 | 1.419 | 1.045 |
| | failure | 0.883 | 1.000 | 0.667 | 0.875 | 0.950 | 0.983 | 0.917 | 0.925 | 0.667 | 0.950 | 0.950 | 0.950 |
| | reward | 0.655 | 0.650 | 0.283 | 0.242 | **0.485** | 0.508 | 0.666 | 0.483 | 0.307 | 0.218 | 0.174 | 0.138 |
| **DT+SAS(ours)** | cost | 1.185 | **1.783** | **0.622** | **0.639** | 1.375 | 1.205 | **0.846** | **1.148** | **0.513** | **1.158** | **1.083** | **0.836** |
| | failure | **0.867** | **0.983** | 0.767 | **0.850** | 0.950 | **0.967** | **0.867** | **0.850** | **0.483** | **0.900** | 0.975 | 1.000 |
| | reward | 0.665 | 0.587 | 0.303 | 0.240 | 0.445 | 0.462 | 0.672 | 0.507 | 0.311 | 0.230 | 0.175 | 0.111 |
| DT+rand | cost | 1.258 | 1.811 | 0.678 | 0.758 | 1.485 | **0.960** | 1.002 | 1.438 | 0.549 | 1.341 | 1.259 | 0.963 |
| | failure | 0.900 | 1.000 | 0.767 | 0.875 | 1.000 | 0.950 | 0.867 | 0.975 | 0.617 | 0.950 | 0.975 | 0.925 |
| | reward | 0.644 | 0.521 | 0.271 | 0.200 | 0.486 | 0.441 | 0.636 | 0.512 | 0.321 | 0.206 | 0.131 | 0.126 |
| DT+maxmax | cost | **0.990** | 2.152 | 0.640 | 0.730 | 1.808 | 1.273 | 1.034 | 1.497 | 0.574 | 1.271 | 1.103 | 0.911 |
| | failure | 0.775 | 1.000 | 0.767 | 0.783 | 1.000 | 0.950 | 0.817 | 0.933 | 0.625 | 0.975 | 0.967 | 0.975 |

Table 2: Full Results in Safety Gymnasium. The values are averaged across three different cost thresholds, 20 evaluation episodes, and three random seeds. Gray: Unsafe agents. **Bold**: Safe agents whose normalized cost is less than 1. **Blue**: Agents which has highest reward among safe agents.

| Task | DT + SAS | | CDT + SAS | | CDT | | BC-All | | BC-Safe | | BCQ-Lag | | BEAR-Lag | | CPQ | | COptiDICE | | DCRL | |
|---|---|---|---|---|---|---|---|---|---|---|---|---|---|---|---|---|---|---|---|---|
| | reward | cost | reward | cost | reward | cost | reward | cost | reward | cost | reward | cost | reward | cost | reward | cost | reward | cost | reward | cost |
| PointGoal1 | 0.66 | 1.19 | 0.65 | 1.27 | 0.69 | 1.12 | **0.65** | 0.95 | 0.43 | 0.54 | **0.71** | 0.98 | 0.74 | 1.18 | 0.57 | **0.35** | 0.49 | 1.66 | 0.24 | 0.86 |
| PointGoal2 | 0.65 | 1.78 | **0.52** | **0.94** | 0.59 | 1.34 | 0.54 | 1.97 | 0.29 | 0.78 | 0.67 | 3.18 | 0.4 | 1.31 | 0.38 | 1.92 | | | 0.28 | 0.26 |
| PointPush1 | **0.28** | **0.62** | 0.26 | 0.54 | 0.24 | 0.48 | 0.19 | 0.61 | 0.13 | 0.43 | **0.33** | 0.86 | 0.22 | 0.79 | 0.2 | 0.83 | 0.13 | 0.83 | 0.01 | 0.52 |
| PointPush2 | **0.24** | **0.64** | 0.20 | 0.53 | 0.21 | 0.65 | 0.18 | 0.91 | 0.11 | 0.8 | 0.23 | 0.99 | 0.16 | 0.89 | 0.11 | 1.04 | 0.02 | 1.18 | 0.02 | 0.07 |
| PointButton1 | 0.49 | 1.38 | 0.51 | 1.27 | 0.5 | 1.68 | 0.1 | 10.5 | **0.06** | **0.52** | 0.24 | 1.73 | 0.2 | 1.6 | 0.69 | 3.2 | 0.13 | 1.4 | 0.01 | 0.48 |
| PointButton2 | 0.51 | 1.14 | **0.41** | **0.98** | 0.46 | 1.57 | 0.27 | 2.02 | 0.16 | 1.1 | 0.4 | 2.66 | 0.43 | 2.47 | 0.58 | 4.3 | 0.15 | 1.51 | 0.18 | 0.64 |
| CarGoal1 | **0.67** | **0.85** | 0.65 | 0.90 | 0.66 | 1.21 | **0.39** | **0.33** | 0.24 | 0.28 | 0.47 | 0.78 | 0.61 | 1.13 | 0.79 | 1.42 | 0.35 | 0.54 | 0.35 | 0.88 |
| CarGoal2 | 0.48 | 1.15 | **0.42** | **0.98** | 0.48 | 1.25 | 0.23 | 1.05 | 0.14 | 0.51 | 0.3 | 1.44 | 0.28 | 1.01 | 0.65 | 3.75 | 0.25 | 0.91 | 0.11 | 2.51 |
| CarPush1 | **0.31** | **0.51** | 0.31 | 0.49 | 0.31 | 0.4 | 0.22 | 0.36 | 0.14 | 0.33 | 0.23 | 0.43 | 0.21 | 0.54 | -0.03 | 0.95 | 0.23 | 0.5 | -0.1 | 0.09 |
| CarPush2 | 0.22 | 1.16 | **0.21** | **0.75** | 0.19 | 1.3 | **0.14** | 0.9 | 0.05 | 0.45 | 0.15 | 1.38 | 0.1 | 1.2 | 0.24 | 4.25 | 0.09 | 1.07 | -0.13 | 0.17 |
| CarButton1 | 0.17 | 1.08 | **0.27** | **0.98** | 0.21 | 1.6 | 0.03 | 1.38 | **0.07** | **0.85** | 0.04 | 1.63 | 0.18 | 2.72 | 0.42 | 9.66 | -0.08 | 1.68 | 0.12 | 0.95 |
| CarButton2 | **0.14** | **0.84** | 0.30 | 1.11 | 0.13 | 1.58 | -0.13 | 1.24 | -0.01 | **0.63** | 0.06 | 2.13 | -0.01 | 2.29 | 0.37 | 12.51 | -0.07 | 1.59 | 0.09 | 1.42 |

insufficiently by evaluating long-horizon $\hat{\rho}$ enough, so SAS can correct long-term behavior that can increase reward, but cost also increase by the absence of enough cost information.

DT+SAS has the lower cost compared to random trajectory instruction (DT+rand) in table 1. The cost values of DT+rand are higher than DT in half of total safety-gymnasium tasks. We can confirm that SAS is a valid self-generated instruction for DT. SAS uses $E$ in Condition $\mathcal{U}_t$ to be more stable and selects the trajectory with the minimum value of the maximum $E$ among steps in a trajectory. For ablation study, we also conduct

Table 3: Performance of the DT and DT+SAS in the MuJoCo environments with D4RL datasets. Only in this table, we compute rewards using the normalized scoring method from the CQL paper (Kumar et al., 2020). **Bold**: Agents with lower failure or higher reward.

| Environment | | Hopper | | Walker2d | | Humanoid | |
|---|---|---|---|---|---|---|---|
| | | expert | medium | expert | medium | expert | medium |
| DT | reward | 110.7 | 86.6 | 107.7 | 82.2 | 98.5 | 40.5 |
| | failure | 0.05 | 1 | 0 | 0.54 | 0.20 | 0.97 |
| **DT+SAS** | reward | 110.7 | **87.5** | 107.7 | **89.5** | **103.5** | **50.6** |
| | failure | **0.03** | 1 | 0 | **0.46** | **0.10** | **0.87** |

the case of selecting a trajectory with the maximum value of maximum step $E$ among trajectories (maxmax). In table 1, it is evident that, compared to the DT+SAS model, DT+maxmax model exhibits higher cost and failure in the majority of environments. Additionally, the DT+maxmax model demonstrates lower reward values compared to the DT+SAS model, except in 2 tasks. As seen in the results of DT+rand and DT+maxmax, our SAS algorithm-based prompting, which verifies the Lyapunov function, enables the Transformer to make much safer choices during the action selection process. This demonstrates that providing a prompt generated before interacting with the actual environment influences the overall performance throughout the episode, much like selecting an appropriate initial skill in hierarchical RL. In table 3, DT was trained on both medium and expert datasets. In mujoco, cost is not explicitly provided, so we only report reward and failure. As we mentioned above, we evaluate the failure when the agent terminated before max episode length. DT+SAS generally shows higher reward and lower failure compared to DT. In Walker2d, the failure rate of DT for expert dataset is already 0 with 100 episodes, so we cannot observe the improvement of SAS. However, for medium dataset, we observe the better performance in both reward and failure,

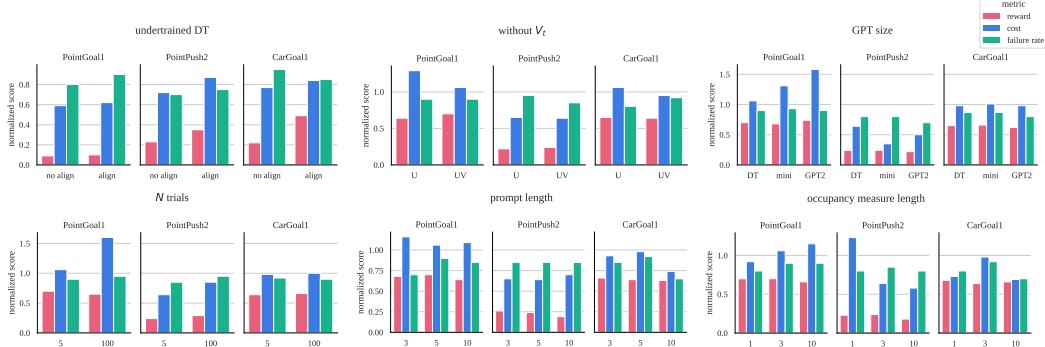

Figure 4: The undertrained DT graph illustrates the performance with and without SAS to the undertrained DT. Without $\mathcal{V}_t$ represents the results of the ablation study without $\mathcal{V}_t$ in section 5.3. UV corresponds to SAS, and U represents without applying $\mathcal{V}_t$. The remaining figures are in Appendix C.

which means SAS is also effective in Walker2d. In Humanoid, both in the expert and medium dataset, DT+SAS outperforms DT in all three metrics. APE-V algorithm (Ghosh et al., 2022a) (belief-based adaptation) uses offline ensemble C51 with SAC-$N$ to enhance the performance by adaptive training for downstream task. In walker2d medium APE-V algorithm improved the average return by 2.7%, but we note that DT+SAS outperform DT by improving 8.9% for the average return in Table 3. SAS does not require fine-tuning or retraining, but APE-V shows the worse test-time performance.

**Does SAS outperform than offline safe RL methods?** Since SAS is designed for test time adaptation of DT, we can apply both DT and CDT for alignment. In table 2, We can see that SAS method shows safer performance than without SAS, as cost and failure rate decrease in most tasks. Except for PG1, PP1, and PB1 environments, DT+SAS or CDT+SAS achieves the highest rewards among all baselines even with cost less than 1. In particular, in PB2, CDT+SAS stands out as the only safe algorithm demonstrating decent rewards. Compared to baselines, CDT+SAS exhibits superior performance in CB2, while in CB1, DT+SAS performs remarkably better. When we compare CDT+SAS with CDT, it is evident that cost consistently decreases. In addition, in six tasks, cost even decreases falling below 1, which means it lowers the target cost to be safe. SAS ensures that, at test time, the pretrained DT can be aligned better with the distribution of the offline dataset. When DT is worse than the collected expert in offline dataset, SAS boost the performance of reward. We also conduct the case that the Decision Transformer that had not been sufficiently trained (undertrained DT), and the outcomes are detailed in fig. 4. As observed, while the cost and failure rates experienced an increase, the reward also increased. Our method is effective in enhancing the reward of a less trained Decision Transformer at the test time. We utilize the initial prompt derived from $\mathcal{U}_t$ and generate the prompt with $\mathcal{V}_t$. We conducted tests using the initial prompt obtained from $\mathcal{U}_t$ directly at test time, without incorporating $\mathcal{V}_t$. We can observe that the cost decreases in all three environments. In the case of failure, the failure decreased in all environments except for the CarGoal1 tasks.

Offline RL methods often rely heavily on one-step RL, whereas our SAS approach performs depth-first search during the inference process through internal thought. This allows for verification of safe control performance for the entire episode of the selected high-level policy. This advantage explains why our method outperforms traditional safe RL methods. It's also important to note that even in scenarios where cost-based offline safe RL has already been applied to CDT, prompting can further improve the overall performance throughout the episode which can be seen in table 2.

## 7 CONCLUSION

Deploying downstream controller with offline RL is an important key to achieving real-world deep RL practical. Unlike the other machine learning domains, such as NLP, it is hard to collect high quality real-world dataset for pretraining. To solve this problem, we propose self-alignment method for transformer based RL to align an offline RL agent to be stable for safety and better performance. It is hoped that the proposed method may trigger new insights on further improvements in safe exploration and stable downstream task deployment in RL.

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

## A   MODEL ARCHITECTURE

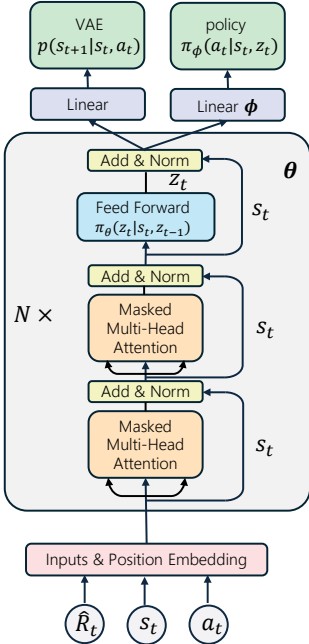

Figure 5: The architecture of decision transformer with VAE for model-based RL. The only difference with decision transformer is the additional linear layer and VAE decoder to predict the next state. We consider the output of feed-forward layer as the predictor of $z_t$ with the parameter $\theta$ which corresponds to the high-level policy, and the combined values of $s_t, z_t$ by the attention and residual connection are fed into the low-level policy with the linear layer $\phi$.

## B   EXPERIMENT SETTING AND HYPERPARAMETERS

### B.1   EXPERIMENT SETTING

We conduct Hopper, Walker2d, and Humanoid in OpenAI Gym, where the agent fails and terminates when the sum of unhealthy rewards get larger. For Safety Gymnasium, we use two different robots (`Point`, `Car`) in 3 tasks (`Goal`, `Push`, `Button`) with two difficulties (1, 2) respectively. In `Goal` and `Button` tasks, an agent navigate to the goal while avoiding touching hazards, and an agent push a box to the goal in `Push` task. We denote normalized reward and cost returns as reward and cost for simplicity, and use failure in Tables for failure rate. If an agent experiences any cost due to encountering a hazard within an episode or exceeding unhealthy cost for mujoco (terminated), we considered that episode as a failure episode. The baselines we used are CDT (Liu et al., 2023b), Imitation Learning (BC-Safe, BC-All(Liu et al., 2023b; Xu et al., 2022a)), Distribution Correction Estimation (COptiDICE(Lee et al., 2022)), and Q-learning (CPQ, BCQ-Lag, BEAR-Lag(Xu et al., 2022a)).

### B.2   NORMALIZED SCORE

We applied normalization to both reward return and cost return to make it easier to compare for all environments. Let $r_{max}(\mathcal{M})$ and $r_{min}(\mathcal{M})$ denote the maximum reward return and minimum reward return in the dataset $\mathcal{T}$, respectively. Then, the normalized reward return is computed as:

$$R_{normalized} = \frac{R_\pi - r_{min}(\mathcal{M}))}{r_{max}(\mathcal{M}) - r_{min}(\mathcal{M})}$$

where $\mathcal{R}_\pi$ denotes the evaluated reward return obtained by the agent. Normalized cost return is defined as the ratio between the cost return obtained by the agent and the target cost $\kappa$:

$$C_{normalized} = \frac{C_\pi + \epsilon}{\kappa + \epsilon}$$

where $\epsilon$ is a small positive number for numerical stability. The values are averaged across three different cost thresholds, 20 evaluation episodes, and three random seeds.

### B.3 DATASET DETAILS

We conducted experiments using the OpenAI Gym's medium and expert datasets from `https://github.com/Farama-Foundation/D4RL` and the Safety Gymnasium's expert dataset from `https://github.com/liuzuxin/OSRL/tree/main`. Detailed information about the dataset is presented in Table 4. The Max Cost means the maximum cost return in dataset trajectories.

Table 4: Dataset details

| Benchmark | Task | Max Timestep | Action Space | State Space | Max Cost | Trajectories |
|---|---|---|---|---|---|---|
| Safety Gymnasium | SafetyPointGoal1-v0 | 1000 | 2 | 60 | 100 | 2022 |
| | SafetyPointGoal2-v0 | | | 60 | 200 | 3442 |
| | SafetyPointPush1-v0 | | | 76 | 150 | 2379 |
| | SafetyPointPush2-v0 | | | 76 | 200 | 3242 |
| | SafetyPointButton1-v0 | | | 76 | 200 | 2268 |
| | SafetyPointButton2-v0 | | | 76 | 250 | 3288 |
| | SafetyCarGoal1-v0 | | | 72 | 200 | 1671 |
| | SafetyCarGoal2-v0 | | | 72 | 250 | 4105 |
| | SafetyCarPush1-v0 | | | 88 | 250 | 2871 |
| | SafetyCarPush2-v0 | | | 88 | 400 | 4407 |
| | SafetyCarButton1-v0 | | | 88 | 250 | 2656 |
| | SafetyCarButton2-v0 | | | 88 | 300 | 3755 |

### B.4 HYPERPARAMETERS FOR THE EXPERIMENTS

During the training of Decision Transformer, we applied warmup for the first 10000 steps, and we used the ReLU activation function. Further details about the hyperparameters can be found in table 5.

Table 5: Hyperparameters for the experiments

| Common Parameters | Safety-Gymnasium | Parameters | CDT | DT |
|---|---|---|---|---|
| Action hidden size | [256, 256] for all methods except CDT, DT | Number of layers | 3 | 3 |
| VAE hidden size | [400, 400] BCQ-Lag, BEAR-Lag, CPQ | Number of attention heads | 8 | 1 |
| Cost thresholds | [20, 40, 80] | Embedding dimension | 128 | 128 |
| Gradient steps | 100000 | Batch size | 2048 | 64 |
| $[K_\mathcal{P}, K_\mathcal{I}, K_\mathcal{D}]$ | [0.1, 0.003, 0.001] BCQ-Lag, BEAR-Lag | Context length K | 300 | 20 |
| Batch size | 512 | Learning rate | 0.0001 | 0.0001 |
| Actor learning rate | 0.0001 | Dropout | 0.1 | 0.1 |
| Critic learning rate | 0.001 | Adam betas | (0.9, 0.999) | (0.9, 0.999) |

## C ABLATION STUDIES

### C.1 NUMBER OF TRAJECTORIES SAMPLED FOR IMAGINATION

We employ Decision Transformer to imagine multiple trajectories under both condition $\mathcal{U}_t$ and condition $\mathcal{V}_t$. In our case, we sampled 5 trajectories for each condition $\mathcal{U}_t$ and $\mathcal{V}_t$. As part of an ablation study, we compared the results of sampling 100 trajectories in experiment with our experimental results. As we can see in fig. 6, the experiment revealed that there was not a significant difference in the model's performance due to the difference in the number of sampled trajectories. In PointGoal1 environment, an increase in cost was observed when the number of sampled trajectories was 100.

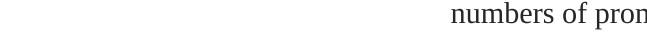

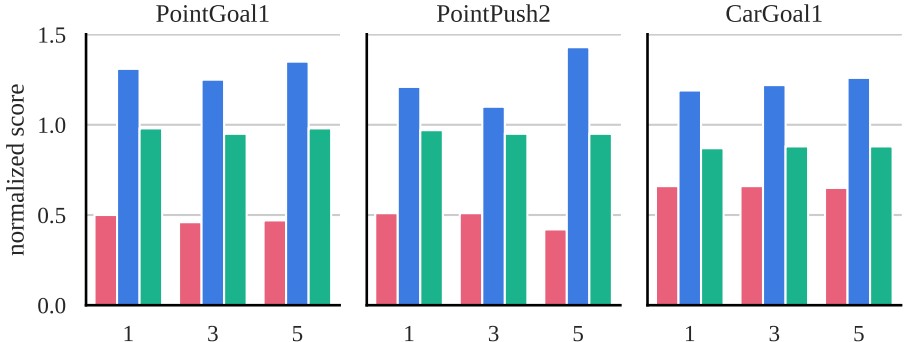

Figure 6: Ablation studies on number of trajectories sampled for imagination. Red bar, blue bar, green bar is reward, cost, failure score respectively

### C.2 TIME STEP LENGTH TO CALCULATE $E$

We calculated and approximated $E$ from trajectories imagined by Decision Transformer under both conditions U and V. We conducted experiments with a default time step length of 3 for computing $E$. As part of an ablation study, we also experimented with time step lengths of 1 and 10, comparing the results with our findings, which are presented in fig. 4. In the results for the CarGoal1 environment, the cost is lowest when the time step length is 10, while in the PointGoal1 environment, it is actually highest. This indicates that increasing the time step length for calculating $E$ does not noticeably improve the model's performance.

### C.3 TIME STEP LENGTH OF TRAJECTORY IN PROMPT

We extracted the trajectory from a specific time t to 5 time steps before that from trajectories generated through Condition $\mathcal{U}_t$ and $\mathcal{V}_t$. We then fed this truncated trajectory into the prompt of Decision Transformer at the test time. As part of an ablation study, we experimented with the time step length of Decision Transformer's prompt, setting it to 3 and 10, and the results are presented in fig. 4. For each time step length of the prompt (3, 5, 10), there are instances where the cost in the experimental results is the highest, as well as instances where it is the lowest. Hence, it can be concluded that the time step length of the prompt does not significantly impact the model's performance.

### C.4 NUMBER OF PROMPTS

We proceeded by using a single trajectory fragment generated by our algorithm as the prompt for Decision Transformer. As part of an ablation study, we compared the performance of our approach with the method of concatenating three or five trajectory fragments obtained by running our algorithm three or five times, respectively, and using them as a prompt. The experimental results in fig. 4 show that the method of using five trajectory fragments as a prompt resulted in higher costs. While there is some difference in the PointPush2 environment when the number of fragments is 1 or 3, overall, the performance fluctuates without a clear trend.

### C.5 MODEL SIZE OF DECISION TRANSFORMER

We conducted experiments to observe how the effectiveness of SAS varies with the model size of the Decision Transformer. Starting from the smallest size, the default Decision Transformer, we experimented with sizes ranging from gpt-mini to larger sizes like gpt2, and the results are depicted in fig. 4. When examining the PointGoal1 environment, it seems that as the model size increases, the cost also tends to increase. However, looking at the PointPush2 environment, the opposite trend is observed, where the model with the smallest size has the highest cost, suggesting that there may not

be a significant correlation. However, concerning failures, except for the gpt-mini size in PointGoal1, it can be observed that as the model size increases, failures generally decrease.

# D  COMPLETE EXPERIMENT RESULTS

## D.1  RESULTS FOR ALL THE DATASETS

We present the results for a total of 16 datasets in table 6. These results include an additional experiment on four Circle tasks (PointCircle1, PointCircle2, CarCircle1, CarCircle2) and eight tasks in bullet-safety-gym environment(BallRun, CarRun, DroneRun, AntRun, BallCircle, CarCircle, DroneCircle, AntCircle). In PC2, CC1, and CC2 environments, CDT+SAS exhibited the highest reward among safe agents. CDT+SAS demonstrates lower costs than CDT in all four environments.

Table 6: Complete evaluation results of the baselines and the Decision Transformer with our method (DT+SAS) and Constrained Decision Transformer with our method (CDT+SAS) in the Safety Gymnasium environment. The values are averaged across three different cost thresholds, 20 evaluation episodes, and three random seeds. Gray: Unsafe agents. **Bold**: Safe agents whose normalized cost is less than 1. **Blue**: Agents which has highest reward among safe agents

| Task | DT + ours | | CDT + ours | | CDT | | BC-All | | BC-Safe | | BCQ-Lag | | BEAR-Lag | | CPQ | | COptiDICE | |
|---|---|---|---|---|---|---|---|---|---|---|---|---|---|---|---|---|---|---|---|
| | reward | cost | reward | cost | reward | cost | reward | cost | reward | cost | reward | cost | reward | cost | reward | cost | reward | cost |
| PointGoal1 | 0.66 | 1.19 | 0.65 | 1.27 | 0.69 | 1.12 | **0.65** | **0.95** | **0.43** | **0.54** | **0.71** | **0.98** | 0.74 | 1.18 | **0.57** | **0.35** | 0.49 | 1.66 |
| PointGoal2 | 0.65 | 1.78 | **0.52** | **0.94** | 0.59 | 1.34 | 0.54 | 1.97 | **0.29** | **0.78** | 0.67 | 3.18 | 0.67 | 3.11 | 0.4 | 1.31 | 0.38 | 1.92 |
| PointPush1 | 0.28 | 0.62 | 0.26 | 0.54 | 0.24 | 0.48 | 0.19 | 0.61 | 0.13 | 0.43 | **0.33** | **0.86** | 0.22 | 0.79 | 0.2 | 0.83 | 0.13 | 0.83 |
| PointPush2 | **0.24** | **0.64** | 0.20 | 0.53 | 0.21 | 0.65 | 0.18 | 0.91 | 0.11 | 0.8 | 0.23 | 0.99 | 0.16 | 0.89 | 0.11 | 1.04 | 0.02 | 1.18 |
| PointButton1 | 0.49 | 1.38 | 0.51 | 1.27 | 0.5 | 1.68 | 0.1 | 10.5 | **0.06** | **0.52** | 0.24 | 1.73 | 0.2 | 1.6 | 0.69 | 3.2 | 0.13 | 1.4 |
| PointButton2 | 0.51 | 1.14 | **0.41** | **0.98** | 0.46 | 1.57 | 0.27 | 2.02 | 0.16 | 1.1 | 0.4 | 2.66 | 0.43 | 2.47 | 0.58 | 4.3 | 0.15 | 1.51 |
| PointCircle1 | 0.69 | 1.81 | **0.54** | **0.21** | 0.59 | 0.69 | 0.79 | 3.98 | **0.41** | **0.16** | 0.54 | 2.38 | 0.73 | 3.28 | **0.43** | **0.75** | 0.86 | 5.51 |
| PointCircle2 | 0.42 | 1.69 | **0.63** | **0.47** | 0.64 | 1.05 | 0.66 | 4.17 | **0.48** | **0.99** | 0.66 | 2.6 | 0.63 | 4.27 | 0.24 | 3.58 | 0.85 | 8.61 |
| CarGoal1 | **0.67** | **0.85** | 0.65 | 0.90 | 0.66 | 1.21 | **0.39** | **0.33** | **0.24** | **0.28** | **0.47** | **0.78** | 0.61 | 1.13 | 0.79 | 1.42 | **0.35** | **0.54** |
| CarGoal2 | 0.48 | 1.15 | **0.42** | **0.98** | 0.48 | 1.25 | 0.23 | 1.05 | 0.14 | 0.51 | 0.3 | 1.44 | 0.28 | 1.01 | 0.65 | 3.75 | **0.25** | **0.91** |
| CarPush1 | **0.31** | **0.51** | 0.31 | 0.49 | 0.31 | 0.4 | 0.22 | 0.36 | 0.14 | 0.33 | 0.23 | 0.43 | 0.21 | 0.54 | -0.03 | 0.95 | 0.23 | 0.5 |
| CarPush2 | 0.22 | 1.16 | **0.21** | **0.75** | 0.19 | 1.3 | **0.14** | **0.9** | 0.05 | 0.45 | 0.15 | 1.38 | 0.1 | 1.2 | 0.24 | 4.25 | 0.09 | 1.07 |
| CarButton1 | 0.17 | 1.08 | **0.27** | **0.98** | 0.21 | 1.6 | 0.03 | 1.38 | 0.07 | 0.85 | 0.04 | 1.63 | 0.18 | 2.72 | 0.42 | 9.66 | -0.08 | 1.68 |
| CarButton2 | **0.14** | **0.84** | 0.30 | 1.11 | 0.13 | 1.58 | -0.13 | 1.24 | **-0.01** | **0.63** | 0.06 | 2.13 | -0.01 | 2.29 | 0.37 | 12.51 | -0.07 | 1.59 |
| CarCircle1 | 0.41 | 1.84 | **0.47** | **0.52** | 0.6 | 1.73 | 0.72 | 4.39 | 0.37 | 1.38 | 0.73 | 5.25 | 0.76 | 5.46 | 0.02 | 2.29 | 0.7 | 5.72 |
| CarCircle2 | 0.63 | 1.69 | **0.56** | **0.62** | 0.66 | 2.53 | 0.76 | 6.44 | 0.54 | 3.38 | 0.72 | 6.58 | 0.74 | 6.82 | 0.44 | 2.69 | 0.77 | 7.99 |
| BallRun | 0.99 | 1.6 | **0.04** | **0.29** | 0.39 | 1.16 | 0.6 | 5.08 | 0.27 | 1.46 | 0.76 | 3.91 | -0.47 | 5.03 | 0.22 | 1.27 | 0.59 | 3.52 |
| CarRun | 8.12 | 1.06 | **0.72** | **0.39** | 0.99 | 0.65 | 0.97 | 0.33 | 0.94 | 0.22 | 0.94 | 0.15 | 0.68 | 7.78 | 0.95 | 1.79 | 0.87 | 0 |
| DroneRun | 0.76 | 1.58 | **0.33** | **0.78** | 0.63 | 0.79 | 0.24 | 2.13 | 0.28 | 0.74 | 0.72 | 5.54 | 0.42 | 2.47 | 0.33 | 3.52 | 0.67 | 4.15 |
| AntRun | 1.08 | 2.43 | **0.32** | **0.14** | 0.72 | 0.91 | 0.72 | 2.93 | 0.65 | 1.09 | 0.76 | 5.11 | **0.15** | **0.73** | 0.03 | 0.02 | 0.61 | 0.94 |
| BallCircle | 0.81 | 1.41 | **0.32** | **0.38** | 0.77 | 1.07 | 0.74 | 4.71 | 0.52 | 0.65 | 0.69 | 2.36 | 0.86 | 3.09 | **0.64** | **0.76** | 0.7 | 2.61 |
| CarCircle | 0.85 | 1.76 | **0.19** | **0.22** | 0.75 | 0.95 | 0.58 | 3.74 | **0.5** | **0.84** | 0.63 | 1.89 | 0.74 | 2.18 | **0.71** | **0.33** | 0.49 | 3.14 |
| DroneCircle | 0.82 | 1.55 | **0.51** | **0.42** | 0.63 | 0.98 | 0.72 | 3.03 | **0.56** | **0.57** | 0.8 | 3.07 | 0.78 | 3.68 | -0.22 | 1.28 | 0.26 | 1.02 |
| AntCircle | 0.59 | 1.18 | **0.26** | **0.34** | 0.54 | 1.78 | 0.58 | 4.9 | **0.4** | **0.96** | 0.58 | 2.87 | 0.65 | 5.48 | **0** | **0** | 0.17 | 5.04 |

Table 7: The modified version of Table 2 with standard deviation across 3 cost thresholds, 20 evaluation episodes, and 3 random seeds.

| Task | CDT | | | | CDT+ours | | | | DT | | | | DT+ours | | | |
|---|---|---|---|---|---|---|---|---|---|---|---|---|---|---|---|---|
| | reward | | cost | | reward | | cost | | reward | | cost | | reward | | cost | |
| | mean | std | mean | std | mean | std | mean | std | mean | std | mean | std | mean | std | mean | std |
| PointGoal1 | 0.69 | 0.007 | 1.12 | 0.037 | 0.65 | 0.007 | 1.27 | 0.062 | 0.66 | 0.02 | 1.32 | 0.31 | 0.66 | 0.03 | 1.19 | 0.15 |
| PointGoal2 | 0.59 | 0.017 | 1.34 | 0.054 | 0.52 | 0.036 | 0.94 | 0.158 | 0.38 | 0.02 | 2.63 | 0.05 | 0.65 | 0.09 | 1.78 | 0.17 |
| PointPush1 | 0.24 | 0.012 | 0.48 | 0.023 | 0.26 | 0.027 | 0.54 | 0.019 | 0.22 | 0.06 | 0.93 | 0.21 | 0.28 | 0.01 | 0.62 | 0.10 |
| PointPush2 | 0.21 | 1.363 | 0.65 | 31.063 | 0.20 | 0.038 | 0.53 | 0.089 | 0.20 | 0.08 | 0.78 | 0.45 | 0.24 | 0.06 | 0.64 | 0.09 |
| PointButton1 | 0.5 | 0.006 | 1.68 | 0.049 | 0.51 | 0.026 | 1.27 | 0.044 | 0.38 | 0.04 | 1.19 | 0.18 | 0.49 | 0.05 | 1.38 | 0.21 |
| PointButton2 | 0.46 | 0.019 | 1.57 | 0.047 | 0.41 | 0.019 | 0.98 | 0.026 | 0.50 | 0.06 | 1.31 | 0.14 | 0.51 | 0.00 | 1.14 | 0.13 |
| CarGoal1 | 0.66 | 0.008 | 1.21 | 0.057 | 0.65 | 0.008 | 0.90 | 0.035 | 0.64 | 0.02 | 0.98 | 0.12 | 0.67 | 0.03 | 0.85 | 0.16 |
| CarGoal2 | 0.48 | 0.032 | 1.25 | 0.095 | 0.42 | 0.032 | 0.98 | 0.047 | 0.51 | 0.04 | 1.47 | 0.32 | 0.48 | 0.03 | 1.15 | 0.20 |
| CarPush1 | 0.31 | 0.018 | 0.4 | 0.068 | 0.31 | 0.018 | 0.49 | 0.097 | 0.35 | 0.07 | 0.68 | 0.22 | 0.31 | 0.01 | 0.51 | 0.15 |
| CarPush2 | 0.19 | 0.022 | 1.3 | 0.081 | 0.21 | 0.023 | 0.75 | 0.120 | 0.20 | 0.03 | 1.17 | 0.26 | 0.22 | 0.01 | 1.16 | 0.26 |
| CarButton1 | 0.21 | 0.014 | 1.6 | 0.106 | 0.27 | 0.081 | 0.98 | 0.006 | 0.24 | 0.04 | 1.42 | 0.04 | 0.17 | 0.03 | 1.08 | 0.17 |
| CarButton2 | 0.13 | 0.031 | 1.58 | 0.034 | 0.30 | 0.009 | 1.11 | 0.025 | 0.21 | 0.04 | 1.05 | 0.21 | 0.14 | 0.03 | 0.84 | 0.08 |

## D.2 Additional comparison with SOTA offline RL methods and offline meta-RL

We note that our DT+SAS which uses the pretrained DT without cost training data outperforms the above SOTA offline safe RL methods. Moreover, we provide the comparison with CQL, SAC-n, and APE-V which is the online (few-shot) adaptation method for the offline RL algorithm in the table below. We note that our method shows the better improvement compared to the reported value of SAC-n $\rightarrow$ APE-V in APE-V paper. However, the target task of offline meta-RL focuses on the adaptation performance when the goal of the target task changes significantly, which differs critically from measuring the generalization performance that is the aim of our paper, making it challenging to conduct additional experiments.

Table 8: Experiment results with CQL algorithm (Kumar et al., 2020) and APE-V algorithm (Ghosh et al., 2022a) in D4RL (Fu et al., 2021b) datasets.

| Task Name | CQL | DT | | DT+ours | | | SAC-N | APE-V | |
|---|---|---|---|---|---|---|---|---|---|
| | reward | reward | failure | reward | failure | improve(%) | reward | reward | improve(%) |
| hopper-medium-expert | 96.9 | 111.8 | 0.1 | 110.4 | 0.05 | -1.25 | 110 | 105.7 | -3.91 |
| hopper-medium-replay | 86.3 | 94.3 | 0 | 97.3 | 0 | 3.18 | 101.8 | 98.5 | -3.24 |
| walker2d-medium-expert | 109.1 | 108.3 | 0 | 107.5 | 0 | -0.74 | 116 | 110 | -5.17 |
| walker2d-medium-replay | 76.8 | 43.9 | 1 | 69.1 | 0.6 | 57.4 | 78.7 | 82.9 | 5.34 |

# E Proof

We first provide technical results in the main paper. We consider MDP as a graphical model, then we can augment the graphical model with an optimality variable $\mathcal{O}_t$, which denotes $\mathbf{1}\left[(\mathbf{s}_t, \mathbf{a}_t) \in C_t\right]$ where $C_t = \{(\mathbf{s}_t, \mathbf{a}_t) | (\mathbf{s}_t, \mathbf{a}_t) \sim \sum_{\mathbf{z}_t, \mathbf{z}_{t-1}} \pi_\phi^{\text{low}}(\mathbf{a}_t | \mathbf{s}_t, \mathbf{z}_t) \pi_{\theta^*}^{\text{high}}(\mathbf{z}_t | \mathbf{s}_{t-1}, \mathbf{z}_{t-1})\}$, the set of all possible state-action pairs with $\theta^*$. In MDP, we can get high rewarded states in some transitions and hope to allocate high weight for high-rewarded trajectories and low weight for suboptimal trajectories. To denote this high rewarded time-step, we use the above optimality variable $\mathcal{O}_t$.

By defining the condition probability of prompt $p_{1:\mathbf{L}}$ given high-level policy $\pi_\theta^{\text{high}}$, we leverage $r_L(\theta)$ to make sure that the well-designed prompt is selected when it is from underlying the safe high-level policy $\pi_{\theta^*}^{\text{high}}$. In details, the length variable $L$ can be composed of two conditions, the length of prompt and the number of prompt. We conduct the ablation study for this condition in fig. 4. We note that we can have high probability of $p(\mathcal{O}_t = 1 | \mathbf{z}_t) = \exp(r(\pi_\theta^{\text{high}}))$ when we provide the most matching prompt $\mathbf{p}^*$ with the underlying $\pi_{\theta^*}^{\text{high}}$.

## E.1 Proof of eq. (13)

To show the derivation, we start from Equation 1,

$$p(\tau | \mathbf{p}_{1:L}, \mathbf{s}_1^{\text{test}}) = \int_\theta p(\tau | \mathbf{p}_{1:L}, \mathbf{s}_1^{\text{test}}, \theta) p(\theta) d\theta.$$

To check the optimality between the generated trajectory and the prompt, we prove the following equation.

$$p(\mathcal{O}_{\text{traj}} | \mathbf{p}_{1:L}, \mathbf{s}_1^{\text{test}}) = \int_\theta \sum_{\mathbf{z}_1^{\text{test}} \in \mathcal{Z}} \left( g_{\pi_\theta}(\tau, \mathbf{z}_1^{\text{test}}) \prod_{t=1} p(\mathcal{O}_t | \mathbf{s}_t^{\text{test}}, \mathbf{a}_t^{\text{test}}) \right) e^{L \cdot r_L(\theta)} p(\theta) d\theta,$$

where

$$\sum_{\mathbf{z}_1^{\text{test}} \in \mathcal{Z}} \prod_{t=1} p(\mathbf{s}_{t+1}^{\text{test}} | \mathbf{s}_t^{\text{test}}, \mathbf{a}_t^{\text{test}}) \underbrace{p(\mathbf{a}_t^{\text{test}} | \mathbf{s}_t^{\text{test}}, \mathbf{z}_t^{\text{test}})}_{\pi_\phi^{\text{low}}} \underbrace{p_\theta(\mathbf{z}_t^{\text{test}} | \mathbf{s}_t^{\text{test}}, \mathbf{z}_{t-1}^{\text{test}})}_{\pi_\theta^{\text{high}}} =: \sum_{\mathbf{z}_1^{\text{test}} \in \mathcal{Z}} g_{\pi_\theta}(\tau, \mathbf{z}_1^{\text{test}})$$

By the Bayes' rule and the law of total probability, we have

$$p(\mathcal{O}_{\text{traj}}|\mathbf{p}_{1:L}, \mathbf{s}_1^{\text{test}}) = \int_\theta p(\tau|\mathbf{p}_{1:L}, \mathbf{s}_1^{\text{test}}, \theta) p(\theta|\mathbf{p}_{1:L}, \mathbf{s}_t^{\text{test}}) d\theta$$

$$\propto \int_\theta p(\tau|\mathbf{p}_{1:L}, \mathbf{s}_1^{\text{test}}, \theta) p(\mathbf{p}_{1:L}, \mathbf{s}_t^{\text{test}}|\theta) p(\theta) d\theta$$

$$= \int_\theta \sum_{\mathbf{z}_1^{\text{test}} \in \mathcal{Z}} \left( g_{\pi_\theta}(\tau, \mathbf{z}_1^{\text{test}}) \prod_{t=1} p(\mathcal{O}_t|\mathbf{s}_t^{\text{test}}, \mathbf{a}_t^{\text{test}}) \right) \frac{p(\mathbf{p}_{1:L}, \mathbf{s}_t^{\text{test}}|\theta)}{p(\mathbf{p}_{1:L}, \mathbf{s}_t^{\text{test}}|\theta^*)} p(\theta) d\theta$$

$$= \int_\theta \sum_{\mathbf{z}_1^{\text{test}} \in \mathcal{Z}} \left( g_{\pi_\theta}(\tau, \mathbf{z}_1^{\text{test}}) \prod_{t=1} p(\mathcal{O}_t|\mathbf{s}_t^{\text{test}}, \mathbf{a}_t^{\text{test}}) \right) \frac{p(\mathcal{O}_{1:L}, \mathbf{s}_t^{\text{test}}|\theta)}{p(\mathcal{O}_{1:L}, \mathbf{s}_t^{\text{test}}|\theta^*)} p(\theta) d\theta$$

$$= \int_\theta \sum_{\mathbf{z}_1^{\text{test}} \in \mathcal{Z}} \left( g_{\pi_\theta}(\tau, \mathbf{z}_1^{\text{test}}) \prod_{t=1} p(\mathcal{O}_t|\mathbf{s}_t^{\text{test}}, \mathbf{a}_t^{\text{test}}) \right) \exp\left( L \cdot r_L(\theta) \right) p(\theta) d\theta.$$

By the definition of $r_L(\theta)$, we can show that under distinguishability for all $\pi_\theta^{\text{high}} \neq \pi_{\theta*}^{\text{high}}$, then $r_L(\theta)$converges to a negative constant, and by letting $L \to \infty$ we have $\exp(r_L(\pi_\theta^{\text{high}})) = 0$ for all $\pi_\theta^{\text{high}} \neq \pi_{\theta*}^{\text{high}}$ and $\exp(r(\pi_\theta^{\text{high}})) = 1$ for $\pi_\theta^{\text{high}} = \pi_{\theta*}^{\text{high}}$. The more detailed derivation of distinguishability is described in (Xie et al., 2021). In addition, we can note that the probability graphical model has the term $p(\mathbf{z}_t|\mathbf{s}_t, \mathbf{z}_{t-1})$, which samples the latent skill variable when $\mathbf{s}_t$ is given. By the definition of high-level policy, we now can call the transformer with latent variables is intrinsically hierarchical RL with high-level policy $\pi_\theta^{\text{high}} = p(\mathbf{z}_t|\mathbf{s}_t, \mathbf{z}_{t-1})$.

As we can explain our transformer as implicit Bayesian inference of in-context learning (Xie et al., 2021), we now have that the safe high-level policy when we successfully sample a trajectory instruction in algorithm 1 to satisfy Lyapunov conditions perfectly in every time step. Then, the in-context learner RL model can also predict action at the given test-time initial state with Lyapunov stable policy.

### E.2 PROOF OF THEOREM 4.1

Since $\mathcal{U}_t$ and $\mathcal{V}_t$ are both optimality variable to indicate their Lyapunov condition, we apply the probability inferecne for RL as follows:

$$\log p(\mathcal{U}_{1:T}, \mathcal{V}_{1:T}|\tau) = \log \left( p(\mathbf{s}_1) \prod_{t=1} p(\mathcal{U}_t, \mathcal{V}_t|\mathbf{s}_t, \mathbf{a}_t) p(\mathbf{s}_{t+1}|\mathbf{s}_t, \mathbf{a}_t) p(\mathbf{a}_t|\mathbf{s}_t, \mathbf{z}_t) p(\mathbf{z}_t|\mathbf{s}_t, \mathbf{z}_{t-1}) \right)$$

$$= \log \left( \prod_{t=1} p(\mathcal{U}_t, \mathcal{V}_t|\mathbf{s}_t, \mathbf{a}_t) \right) + \log \left( \prod_{t=1} p(\mathbf{s}_1) p(\mathbf{s}_{t+1}|\mathbf{s}_t, \mathbf{a}_t) p(\mathbf{a}_t|\mathbf{s}_t, \mathbf{z}_t) p(\mathbf{z}_t|\mathbf{s}_t, \mathbf{z}_{t-1}) \right)$$

$$= \sum_{t=1} \log \left( p(\mathcal{U}_t, \mathcal{V}_t|\mathbf{s}_t, \mathbf{a}_t) \right) + \log \left( \prod_{t=1} p(\mathbf{s}_1) p(\mathbf{s}_{t+1}|\mathbf{s}_t, \mathbf{a}_t) p(\mathbf{a}_t|\mathbf{s}_t, \mathbf{z}_t) p(\mathbf{z}_t|\mathbf{s}_t, \mathbf{z}_{t-1}) \right)$$

$$= \sum_{t=1} \log \left( p(\mathcal{U}_t, \mathcal{V}_t|\mathbf{s}_t, \mathbf{a}_t) \right) + C.$$

When all $\mathcal{U}_t, \mathcal{V}_t$ are 1, then we know that the trajectory gurantess the Lyapunov condition perfectly. Recall that the trajectory is asymptotically stable if the following conditions are satisfied as described in Definition 3.1.

**(1)** $G(\mathbf{s}_e, \mathbf{a}_e) = 0$, **(2)** $G(\mathbf{s}_t, \mathbf{a}_t) > 0$, $\forall(\mathbf{s}_t, \mathbf{a}_t) \neq (\mathbf{s}_e, \mathbf{a}_e)$, **(3)** $G(\mathbf{s}_t, \mathbf{a}_t) \geq G(\mathbf{s}_{t+1}, \mathbf{a}_{t+1})$.

Since we design our Lyapunov function $G_{\text{SAS}}$ as

$$G_{\text{SAS}}(\mathbf{s}_t, \mathbf{a}_t) = \min_\pi \max_{t'} E(\mathbf{s}_{t'}, \pi(\mathbf{s}_{t'})) - E(\mathbf{s}_t, \mathbf{a}_t),$$

the equilibrium point is defined as $G_{\text{SAS}}(s_e, a_e) = \min_\pi \max_{t'} E(s_e, a_e) - E(s_e, a_e) = \hat{E}_j - E(s_e, a_e) = 0$. Then, the condition $\mathcal{U}_t = 1$ corresponds to the condition (2): $G(s_t, a_t) > 0, \forall(\mathbf{s}_t, \mathbf{a}_t) \neq (\mathbf{s}_e, \mathbf{a}_e)$, and the condition $\mathcal{V}_t = 1$ corresponds to $G(\mathbf{s}_t, \mathbf{a}_t) \geq G(\mathbf{s}_{t+1}, \mathbf{a}_{t+1})$.

If we choose the distributions of $\mathcal{U}_t, \mathcal{V}_t$ as

$$p(\mathcal{U}_t = 1|s_t, a_t) \propto \exp\left(\mathbf{1}\left[G_{\text{SAS}}(s_t, a_t) > 0\right]\right),$$
$$p(\mathcal{V}_t = 1|s_t, a_t) \propto \exp\left(\mathbf{1}\left[G_{\text{SAS}}(s_t, a_t) - G_{\text{SAS}}(s_{t+1}, a_{t+1}) \geq 0\right]\right),$$

then, we can rewrite the above equation as

$$\sum_{t=1} \log p(\mathcal{U}_t, \mathcal{V}_t|\mathbf{s}_t, \mathbf{a}_t) = \sum_{t=1} \log p(\mathcal{V}_t|\mathbf{s}_t, \mathbf{a}_t, \mathcal{U}_t)p(\mathcal{U}_t|\mathbf{s}_t, \mathbf{a}_t)$$
$$\propto \sum_{t=1} \mathbf{1}\left[G_{\text{SAS}}(s_t, a_t) > 0\right] + \sum_{t=1} \mathbf{1}\left[G_{\text{SAS}}(s_t, a_t) - G_{\text{SAS}}(s_{t+1}, a_{t+1}) \geq 0\right]$$

Then, the maximizing the above equation implies that the trajectory get close to the Lyapunov condition.

### E.3 Proof of Eq. (10)

The goal of our method is to keep occupancy measures in the distribution of the target control-invariant set $\mathcal{R} = \{(s_t, a_t)|c_1 \leq E(s_t, a_t) \leq c_2\}$ where $E(s_t, a_t) = -\log \rho(s_t, a_t)$ for utilizing the pretrained expert distribution. As our Lyapunov function approximation is defined as

$$G(s_t, a_t) = \min_{i=1,\cdots,N} \max_{j=1,\cdots T} E(s_j, \pi_i(s_j)) - E(s_t, a_t)$$

for $N$ sample trajectories with the episode length $T$ in the first loop of Algorithm 1. Suppose that $c_2$ is some constant that is larger than $\min_{i=1,\cdots,N} \max_{j=1,\cdots T} E(s_j, \pi_i(s_j))$ for any $N, T$. We now demonstrate that Algorithm 1 reduces the probability of escaping from the control invariant set as the numbers of iterations, $N$ and $M$ for the first and second loops, respectively, increase.

**Assumption E.1.** The difference $\|G(s_t, a_t) - G(s_{t+1}, a_{t+1})\|$ in Eq. (3) over the transition $\mathcal{T}$ is bounded as $\|G(s_t, a_t) - G(s_{t+1}, a_{t+1})\| \leq L$ for all $t$.

*Proof.* First note that $P\left[\tau \not\subset \mathcal{R}\right]$ is less than the sum of the probability of $E(s_t, a_t) \geq c_2$ for all data points in $N$ trajectories and the probability that all $M$ trials moves below $E(s_t, a_t) \leq c_1$. By using Markov's inequality for the first term of RHS and Hoeffding's inequality for the second term of RHS. Then, we have

$$P\left[\tau_{\text{best}} \not\subset \mathcal{R}\right] \leq \left(P\left[E(s, a) \geq c_2\right]\right)^{NT} + \left(P\left[\sum_{t=1}^{T} \mathbf{1}(\mathcal{V}_t \neq 1) \geq \frac{\kappa(c_2 - c_1)}{L}\right]\right)^M$$
$$\leq \left[\frac{\mathbb{E}_{(s,a)\sim\mathcal{D}}[E(s, a)]}{c_2}\right]^{NT} + \exp\left(-\frac{2M\kappa^2(c_2 - c_1)^2}{TL^2}\right)$$

for some constant $\kappa$ to describe the average distance to escape. $\qquad\square$

