# OpenReview forum: "Self-Alignment for Offline Safe Reinforcement Learning"
_ICLR.cc/2025/Conference — Submitted to ICLR 2025_

### Official Review · Reviewer_A6MS · 2024-10-31

**Soundness:** 3
**Presentation:** 3
**Contribution:** 2
**Rating:** 6
**Confidence:** 4

**Summary:**

The author present a self-alignment technique by self-generated prompt to guarantee the better safety. The self-generated prompt for safety is based on Lyapunov condition.  To implement selfalignment for safety, author present a novel formulation of Lyapunov condition as a probabilistic inference  and transformer-based RL world model as a model-based hierarchical RL agent, respectively, to  provide in-context learning based self-alignment.

**Strengths:**

In this work, the author presents a simple model-based RL method with a transformer and a world model and proposes a Lyapunov-conditioned self-alignment method, which does not require retraining and conducts the test-time adaptation for the desired criteria. The author shows that the model-based RL with the transformer architecture can be described as a model-based hierarchical RL. As a result,  the author can combine hierarchical RL and in-context learning for self-alignment in transformers. The proposed self-alignment framework aims to make the agent safe by self-instructing with the Lyapunov condition. In experiments,  the author demonstrates that the self-alignment algorithm outperforms safe RL methods in continuous control and safe RL benchmark environments in terms of return, costs,  and failure rate.
This paper is largely well-organized and clear in its presentation.

**Weaknesses:**

The paper introduces the SAS framework, which is built on promising theoretical foundations. However, its evaluation of practical applicability is somewhat limited. For instance, in Figure 3, the experiments focus solely on benchmarking environments like Safety Gymnasium and Mujoco. While these environments are commonly used, they do not fully capture the complexity and uncertainty of real-world scenarios. Specific scenarios such as autonomous driving simulations, dynamic obstacle avoidance in robotic systems, or high-variability logistics tasks (where demand changes unexpectedly) would offer greater insight.

Additionally, the paper does not provide information on the computational cost of this framework, particularly regarding how it scales with more complex environments or higher-dimensional tasks.  The SAS framework relies on self-generated prompts based on existing data, but the paper fails to discuss how well SAS adapts to environments that experience unexpected changes or previously unseen safety hazards.

**Questions:**

1. Could the authors elaborate on how SAS may generalize compared to non-Lyapunov methods?
2. Additionally, could the authors provide specific metrics on computational costs, such as training time, inference time, or memory usage, and how these scale with environment complexity or dimensionality?
3. Lastly, do the authors discuss or demonstrate how SAS performs in scenarios with distribution shifts or novel hazards not present in the training data?

---

> ### Author Response · Authors · 2024-11-27
>
> Thank you for the positive and constructive feedbak for improving our work. Below, we provide the detailed answers of your concern.
>
> > Additionally, the paper does not provide information on the computational cost of this framework, particularly regarding how it scales with more complex environments or higher-dimensional tasks.  Additionally, could the authors provide specific metrics on computational costs, such as training time, inference time, or memory usage, and how these scale with environment complexity or dimensionality?
> >
>
> If the target model is the model-based RL framework with world model, we does not require any additional architecture. However, we add VAE decoder for predicting the next states to DT and CDT, and the pretraining times are slightly increased than the vanilla DT and CDT.
>
> In addition, the wall-clock time for the entire test-time deployment should be approximately 11 times larger than the original DT as we use 10 total iterations for imagination trajectory. We also note that the execution times of imagined episodes and the interaction with real-environment are similar, because the inference procedure and architecute are identical.
>
> We provide the empirical time result on Hopper-medium-expert in the below table. As we mentioned above, the pretraining time is slightly larger than vanilla DT, and SAS uses 32.44 seconds for self-instruction by imagination before starting test-time deployment.
>
> |  | DT | SAS with DT+VAE |
> | --- | --- | --- |
> | trainig time | 9960 sec | 10027 sec |
> | test time(20 iters) | 3.87 sec | 36.31 sec |
>
> > Could the authors elaborate on how SAS may generalize compared to non-Lyapunov methods?
> >
>
> The key motivation of SAS is to prevent the agent from selecting the actions that are more likely to induce the out-of-distribution states in the future. Our Lyapunov-stable policy is designed to keep the agent within the control invariant set. The agent aligned by our method, SAS selects the action with the lowest likelihood of becoming out-of-distribution among the currently available actions in anticipation of future distributional shifts.
>
> The **out-of-distribution** states (samples) are less frequently visited in the offline RL pretraining dataset. The prior offline safe RL agents could select myopic actions that might not have an immediate effect but are more likely to visit these unpredictable out-of-distribution states (samples) in the future than our method due to the overestimation problem with myopic actions.
>
> By preventing the offline RL agent from selecting myopic actions, our alignment method reduces the likelihood of the agent visiting overestimated states during testing.
>
> This approach empirically results in improved reward/cost performance in the Safety-Gymnasium and better reward/failure performance in MuJoCo with the offline RL dataset D4RL, as our method successfully keeps the agent within the in-distribution of the pretrained expert distribution.
>
> > Lastly, do the authors discuss or demonstrate how SAS performs in scenarios with distribution shifts or novel hazards not present in the training data? The SAS framework relies on self-generated prompts based on existing data, but the paper fails to discuss how well SAS adapts to environments that experience unexpected changes or previously unseen safety hazards
> >
>
> The conducted experiments have already covered the case on your concern. Safety-gymnasium environments randomize the positions and angles of hazards and the starting position. Then, the agent is very unlikely to face the identical envrionment settings in the dataset. In addition, vase, the skyblue object in Figure 3 of PointGoal1 is the movable obstacle. This movable objects often occurs the unexpected changes, because the agent can push and move this vase with the cumulative high cost. Even if SAS generates the prompt based on the pretrained distribution as you pointed out, it is still useful to dodge the very low probable states in terms of occupancy measure because the trained occupancy measure describes the chance to encounter a specific observation (placement).
>
> For example, when starting from a point with a similar field of view, the hazards that can be encountered are likely to be similar, so avoiding dangerous actions from this similar view will allow you to avoid potential obstacles in the future with higher probability.

---

> > ### Comment · Reviewer_A6MS · 2024-11-28
> > **Thanks for your response!**
> >
> > Thank you for addressing my questions. I have no further concerns and keep the 6 rating.

---

### Official Review · Reviewer_4mTU · 2024-11-03

**Soundness:** 2
**Presentation:** 2
**Contribution:** 3
**Rating:** 5
**Confidence:** 3

**Summary:**

The authors present a simple model-based RL method with a transformer and a world model, and propose a Lyapunov-conditioned self-alignment method, which does not require retraining and conducts the test-time adaptation for the desired criteria.

**Strengths:**

The idea of prompting a world model using self-generated trajectories is interesting and promising. Experimental results also show that the idea indeed improves the agent's test-time performance in some tasks.

**Weaknesses:**

1. The writing needs to be polished further.

2. Please use the correct citing format.

3. The Lyapunov condition does not seem to be the main contribution of this work, but the authors use a large part of the content to describe it, which misleads readers to correctly judge the novelty of this work.

4. It is unclear what connection between the proposed Self-Alignment and the original Self-Alignment that is used in LLMs is.

5. While the results show the effectiveness of SAS, experiments are conducted in several relatively easy safe RL benchmarks.

**Questions:**

It is better to shorten the method section and highlight the main contribution of your work.

---

> ### Author Response · Authors · 2024-11-27
>
> Thank you for your detailed and constructive feedback. We revise our paper and upload the modified version by using your feedbacks on writing. Below, we present the detailed answers of your concerns. Please let us know if there is any additional concerns.
>
> > The Lyapunov condition does not seem to be the main contribution of this work.
> >
>
> Lyapunov condition itself is not our contribution, but extending Lyapunov condition into the probabilistic inference problem of offline safe RL is one of our main contribution.
>
> - By constructing Equation (4) and (7), we can indirectly reduce the cost in the test-time deployment by defining the control invariant set to keep in the high probable and safe region.
> - By defining a nonparametric Lyapunov function in Equation (6), we can simply estimate the Lyapunov condition during imagination.
> - Theorem 4.1, which is the theoretical result that allows the proposed probabilistic inference problem to verify Lypunov stability on density, leads to generate the imagined instruction prompt for the in-context learning in Equation (13). This in-context learning makes the offline RL agent find the matched skill $z_1^{test}$ to the instruction prompt policy from the pretrained distribution and execute actions with the selected skill $z_1^{test}$.
>
> > It is unclear what connection between the proposed Self-Alignment and the original Self-Alignment that is used in LLMs is.
> >
>
> Section 5.2 present the connection between the proposed self-alginment method and the original method in LLM. We followed the key concept in the self-align process of Dromedary (Sun et al.,2023) for LLMS. Since the objective of Dromedary is to enable the AI model to generate fitting responses that adhere to the established principles, while simultaneously
> minimizing human supervision, we desgin the principle for safe deployment as the Lypunov condition with the occupancy measure, which has 3 principle conditions in Definition 3.1.
>
> Unlike the predefined princple sentence of Dromedary, we generate imagined trajectories by the predefined Algorithm 1. Then, we feed the generated trajectories as internal thought to guide the test-time deployment of the agent.
>
> > While the results show the effectiveness of SAS, experiments are conducted in several relatively easy safe RL benchmarks.
> >
>
> We respectfully disagree your comment, because the state-of-the-art offline safe RL algorithms, such as CDT has not outperformed the expert in more than 9 environment in terms of cost by showing the normalized scores are larger than 1. Since safety-gymnasium environment is POMDP with the limited front view LIDAR inputs, which makes the agent navigate harder, such as deepmind lab environements. In addition, we have demonstrated the performance on the failure rate in mujoco in Table 3 and 8.

---

### Official Review · Reviewer_8v92 · 2024-11-04

**Soundness:** 2
**Presentation:** 2
**Contribution:** 3
**Rating:** 3
**Confidence:** 3

**Summary:**

The author's present a method for self-alignment of offline transformer RL policies using Lyapunov Density Models. The method is an inference-time procedure for online transfer of an offline-trained policy. The algorithm assumes a trained decision transformer architecture, modified to also predict next state and to include a VAE, so that it can be used to generate imagined trajectories at test time. At inference time, this modified DT is used to rollout imagined trajectories conditioned on the current state. The imagined trajectories are then evaluated based on the Lyapunov stability condition and the least violating trajectory is selected. This trajectory is then used as a prompt to the DT model in order to produce a prompt-conditioned action to be executed in the environment. The author test their method on several Safety-Gym and Mujoco environments and compare to vanilla decision transformer and previous Safe RL methods

**Strengths:**

- The motivation for this work is  strong - being able to train policies entirely offline that are then able to be deployed with safety guarantees in the real environment would signigifcantly advance the applicability of RL to real-world scenarios
- The novelty is good - this is a new method which combines DT with Lyapunov conditioning
- The authors compare against many baseline methods on several different tasks

**Weaknesses:**

Major issues:
- I find several parts of the method hard to follow:
   - The training procedure for the DT+VAE model is not provided
   - on line 245-256 the authors mention "sampling policies" but I do not see where policies are sampled in Alg. 1
  - It is not clear how the results in Section 4 are used in Alg. 1
  - It is not clear how the maximization in equation 8 is performed. This seems to correspond to the two loops in Alg 1, but does this mean that U and V are maximized individually?
  - What do $\theta$ and $\psi$ correspond to in the architecture? These are described as parameters of high and low level policies, but there only seems to be one transformer model in the archtiecture.
- I am concerned about the rigor of the theoretical results:
   - Theorem 4.1 "The problem of finding a trajectory from Lyapunov stable controller is equivalent to solve the following
inference problem" refers to a proof in D2, but the proof ends with "Then, the maximizing the above equation implies that the trajectory get close to the Lyapunov condition". First this claim is not proven but also it doesn't match the claim of 4.1 (equivalence)
  - Theorem 4.3 - The proof in the appdenix is only two lines and starts with an equation that needs greater explanation
- The experimental results lack statistical analysis and hence are not convincing. The results presented in Tables 1 & 2 are very hard to read and do not present a robust statistical analysis. No confidence intervals are included and in many cases it appears that the proposed method only improves over the baseline by very slight margins. Without confidence intervals, it is impossible to say if these results show a statistically significant improvement. Moreover, the experiments were only conducted over thee seeds which is quite small. Additional, in Fig 3. The unsafe regions do not appear to align particularly well with the hazards
- The connection between this method and Safe RL does not feel substantive, since this is primarily a method of constraining an offline policy to the data distribution (which many Offline RL algorithms seek to do). The entire section 4 seems to primarily serve to motivate the threshold for the target control invariant set, however it introduces a margin hyperparameter D which essentially makes the threshold a tunable parameter. Is the derivation in Sec 4 necessary? Also, I do not see in Algorithm 1 where this threshold is used, and there is no discussion of how the hyperparameter D is chosen.
- Again regarding the connection to Safe RL - the authors primarily compare their method to Offline Safe RL algorithms, but this does not necessarily seem like the best baselines, since these methods also use the cost in the offline dataset directly to train the algorithm, while this method does not. Would it not make more sense to compare to other offline methods that seek to constrain the policy within-distribution like CQL? A critical difference between the safe RL methods and the conservative RL methods is that Safe RL methods could avoid constraints even if the expert data is sub-optimal, wheres the conservative methods would rely on an assumption that the training data itself is mostly safe and hence staying in distribution results in safety.


Minor issues:
- Citation for Def 3.1
- Several acronyms and variables are not defined, eg.:
   - Eq. 2 $B^c$ doesn’t seem to be defined
   - CDT is only cited in the Appendix but referenced many times in the main text - no where in the text is a description of the method provided.
- There are many very long paragraphs that combine multiple ideas that should be split. For example on the last page, third to last paragraph.
- There are several grammatical errors

**Questions:**

- Decision transformer is reward conditioned - how is the reward conditioning handled in your method?

---

> ### Author Response · Authors · 2024-11-27
> **We thank the reviewer for the detailed review and constructive feedback**
>
> We thank the reviewer for the detailed review and constructive feedback about this work. Below, we present detailed answers of your concerns and provide summarized revision details in the newly uploade version of our paper. We welcome additional discussion and suggestions for improving our work further.
>
> > Q1: The training procedure for the DT+VAE model is not provided. What do $\theta$ and $\phi$ correspond to in the architecture? These are described as parameters of high and low level policies, but there only seems to be one transformer model in the archtiecture.
> >
>
> We have already addressed the training procedure of our DT+VAE model in Figure 5 in Appendix B.4.
>
> For clarity, we upload the additional figure to explain the connection of transformer model and the parameters $\theta, \phi$. Since the motivation of our paper is to apply self-alignment (in-context learning) into the model-based RL with transformer, we only add the VAE decoder on decision transformer as described in Figure 5. We note that $\theta$ corresponds to all parameters of decision transformer except for the last linear layer. The transformer includes the residual connection, so we can consider the last Add & Norm layer as $f(s_t,z_t)$ where $f$ indicates the add & norm operator. Then, $\phi \circ f(s_t,z_t)$ outputs $a_t$, and the last linear layer $\phi$ corresponds to the low-level policy parameter.
>
> > Q2: On lines 245-256, the authors mention "sampling policies," but I do not see where policies are sampled in Alg. 1.
> >
>
> The phrasing in the lines 245-246 was not clearly constructed due to the omission of specific details, which could lead to misunderstanding. To address this, we revise the sentences as follows and highlight the revised paragraphs in red in the uploaded revision.
>
> We note that the first Lyapunov condition observable $U_t$ is indirectly computed by lines 1 to 6 in Algorithm 1. In the first loop of Algorithm 1, among the $N$ iterations, we select the episode with the lowest maximum energy value reached by each imagined trajectory. In line 6, we set the selected lowest maximum energy $ E_j$ as the value of our approximate Lyapunov model for the equilibrium point, $G_{SAS}(s_e,a_e) = \min_\pi \max_{t'} E(s_e,a_e)-E(s_e,a_e)=E_j-E(s_e,a_e)=0$.
> Then, all other $N-1$ episodes have steps with $G(s,a)<0$ inevitably due to $G_{SAS}(s_e,a_e)$, leading to violation of the condition $U_t=1$.
>
> To search for a Lyapunov stable policy which guarantees all state-action pair elements are in $R_G^\textnormal{SAS}$ in Equation (7), we assume that a test-time agent can access a set of previously learned policies, $\Pi=\{\pi_i\}_{i=1}^N$ from pretrained distribution. Each generated trajectory at $i$-th iteration, $\tau_i$, corresponds one-to-one to a certain $\pi_i \in \Pi$ at given initial state $s_0$. As we increase the number of iterations of the first loop, $N \to \infty$, we can get a lower $\hat{E}_j$, which leads to the more probable subsequent $(s,a)$ and equilibrium point.
>
> Furthermore, selecting the most probable index $k^*$ in the second for-loop implies that we choose the optimal-selection policy which violates the condition $V_t$ least.
> Then, the selected $\pi$ is highly likely to satisfying in the control invariant set $[(s,a) | 0 \leq G_{SAS}(s,a) \leq \hat{E}_j  ]$.
>
> > Q3: Decision transformer is reward conditioned - how is the reward conditioning handled in your method?
> >
>
> For the cost threshold, we have already reported 3 different thresholds in Talbe 5. We use the expert return data in the offline RL dataset as reward condition, as DT and CDT have conducted.
>
> > Q4: The experimental results lack statistical analysis and hence are not convincing
> >
>
> We respectfully disagree with your concern, because all our reported values are followed by the existing **offline safe RL experiment protocol**, which is originated from the offline RL benchmark such as D4RL. We note that our experiment is conducted with 3 different cost thresolds, 20 evaluation episodes, and 3 different seed. It means that we **test 180 episodes** for a single expectation value.
>
> Since the reported value is the normalized score over the oracle performance, for example, the change of 0.31 in the reward value for CarGoal1 between CDT and CDT+SAS implies 31% improvement. Then, it is hard to consider our improvement as a slight margin. Furthermore, our method does not use any additional fine-tuning. It is noteworthy that our self-alignment method is useful for test-time adaptation.
>
> In addition, we provide the standard deviation values over 180 tirlas on Table 2 for our methods to help for the better verification in Table 7 (appendix in the revised version).

---

> ### Author Response · Authors · 2024-11-27
>
> > Q5: It is not clear how the results in Section 4 are used in Alg. 1. It is not clear how the maximization in equation 8 is performed.
> >
>
> We first note that the label number of Equation 8 is changed to Equation 9 in the revised version since we add a label for $R_G^\textnormal{SAS}$ as Equation (7). The relationship between Section 4 and Algorithm 1 can be explained in the following three parts.
>
> **the selected lowest maximum  energy $\hat{E}_j$**
>
> As we addressed in Q2, the selected $\hat{E}_j$ in line 6 of Algorithm 1 has a role to compute the condition $U_t$. As we increase $N,$ we can get  a lower maximum energy $\hat{E}_j$ and lead to a tighter control invariant set which forces the agent to execute the highly probable actions with low $E(s_t, a_t)$ .
>
> **how the maximization in equation 8 (changed to 9) is performed**
>
> First, we remind that the first loop to find $\hat{E}_j$in line 6 of Algorithm 1 is to compute $U_t$ over $N$trajectories. To find the trajectory $\tau$ that maximizes Equation 9, we select the episode$j$with the lowest maximum energy $\hat{E}_j$, which has $U_t=1$for all steps $1\leq t \leq T$ over $N$episodes in the first loop.
>
> As we provided in lines 7 to 14 in Algorithm 1, we utilize the in-context learning property in Equation (13) with the demonstration prompt $p_{1:L}$ in line 7 to generate trajectories again for estimating $V_t$ by inheriting the property of the above episode $j$ of which policy demonstrates $U_t=1$ for all steps. Across $M$ episodes from the second loop, we choose the episode (trajectory) which satisfies $V_t=1$ most for $T$ steps. We note that we can estimate $p(V_t=1|U_t=1, \tau)$ since we condition the demonstration prompt $\tilde{p}_{1:L}$ to generate trajectories in the second loop.
>
> **does this mean that $U_t$ and $V_t$ are maximized individually?**
>
> No, we find the trajectory that maximizes the number of steps that has$V_t=1$with the condition from the demonstration prompt that has already achieved the low maximum energy $\hat{E}_j$. With this condition, the imagination process can find the trajectory where the energy does not increase over steps by checking $V_t=1$. It induces that the agent stays in the control invariant set and does not execute an action with higher energy (lower occupancy measure).
>
> > Q6: the rigor of the theoretical results
> >
>
> Thank you for your detailed comments. We add more detailed explanation in the revised version and provide a brief explanation as follows.
>
> **Theorem 4.1**
>
> To address your concern, we provide a simple explanation to show the equivalence of two statements. To connect the inference problem and Lyapunov control, we choose the distribution of $U_t,V_t$ as follows:
>
> $$
> p(U_t=1 | s_t,a_t) \propto \exp \left( \mathbf{1} \left[ G_{SAS}(s_t,a_t) >0 \right] \right)
> $$
> $$
> p(V_t=1 | s_t,a_t) \propto \exp \left( \mathbf{1} \left[ G_{SAS}(s_t,a_t) - G_{SAS}(s_{t+1},a_{t+1}) \geq 0 \right] \right)
> $$
>
> Then, we can have
>
> $$
> \log p(U_t=1 | s_t,a_t) \propto  \mathbf{1} \left[ G_\textnormal{SAS}(s_t,a_t) >0 \right]  \\
> \log p(V_t=1 | s_t,a_t) \propto  \mathbf{1} \left[ G_\textnormal{SAS}(s_t,a_t) - G_\textnormal{SAS}(s_{t+1},a_{t+1}) \geq 0 \right]
> $$
>
> By using the relation, we can simply show the equivalance in the proof of the revised version.
>
> **Proposition 4.3**
>
> The goal of our method is to keep occupancy measures in the distribution of the target control-invariant set $R=\{ (s_t,a_t) | c_1 \leq E(s_t, a_t) \leq c_2\}$ where $E(s_t,a_t) = - \log \rho(s_t,a_t)$ for utilizing the pretrained expert distribution. As our Lyapunov function approximation is defined as
> $G(s_t,a_t) = \underset{i=1,\cdots,N}{\min} \underset{j=1,\cdots T}{\max} E(s_j, \pi_i(s_j)) - E(s_t,a_t)$ for $N$ sample trajectories with the episode length $T$ in the first loop of Algorithm 1.
>
> Suppose that $c_2$ is some constant that is larger than $\underset{i=1,\cdots,N}{\min} \underset{j=1,\cdots T}{\max} E(s_j, \pi_i(s_j))$ for any $N,T$. We now demonstrate that Algorithm 1 reduces the probability of escaping from the control invariant set as the numbers of iterations, $N$ and $M$ for the first and second loops, respectively, increase.
>
> By the definition of Lyapunov stability, it is important to search a state-action pair $(s,a)$ such that $E(s,a)\leq c_2$, because we can stay the searched sate-action pair $(s,a)$ over steps if we find this pair. Then, we cannot have a Lyapunov stable trajectory if all pairs have $E(s,a) \geq c_2$ over $NT$ i.i.d. samplings. Secondly, all episodes always has an pair such that $E(s,a)<c_1$ if all $M$ episodes have more than $\kappa(c_2-c_1) \over L$ decreasing steps during $T$ steps.
>
> Then, we can upper-bound the probability $p(\tau_{best} \not\subset R)$ as the sum of two above probabilities.
>
> By using Markov’s inequality and Hoeffding’s inequality for each base probability respectively, we can have the result of proposition 4.3.

---

> > ### Author Response · Authors · 2024-11-27
> >
> > > Q7: in Fig 3. the unsafe regions do not appear to align particularly well with the hazards
> > >
> >
> > The unsafe region does not necessarily need to align exactly with the areas surrounding hazards. By the definition of Lyapunov stability and control invariant set, it is highly likely to reach an hazard in the offline dataset when the agent enters into the unsafe region.
> >
> > For example, in the upper row case, the agent is highly likely to reach the lower right hazards if the agent enters into the right side unsafe region which is almost empty. This is because the agent suffers from finding the goal point and has no information to act correctly from the dataset.
> >
> > > Q8: Is the derivation in Sec 4 necessary? Also, I do not see in Algorithm 1 where this threshold is used, and there is no discussion of how the hyperparameter D is chosen.
> > >
> >
> > The reason that we introduce the variable $D$ is to generalize our method to an arbitrary offline dataset, which covers from the expert data to the random dataset. As $D$ becomes larger, the worst case in the dataset implies that there exists state-action pairs which is very unlikely to be excuted by the optimal policy. In this case, we should mitigate the upper bound of control invariant set to search a Lyapunov stable trajectory in the larger region.
> >
> > In Algorithm 1, $D$ can be adapted by choosing the size of $N$, since the larger $N$ induces the lower maximum energy $\hat{E}_j$.
> >
> > > Q9: Using the cost information, the comparison with standard offline RL, and the more explanation on the difference between safe RL and our method.
> > >
> >
> > - Our method, CDT+SAS, has already used the cost value directly when we pretrain the CDT networks with the offline safe RL dataset. In test-time, our self-alignment mehtod, SAS, utilizes the occupancy measure of CDT networks to estimate the  cost information which can lead to a trajectory with high costs **as described in Equation (3).**
> > - We also report the performance on the sandard offline RL benchmarks in Table 3 and Table 8. First, Table 3 deomonstrates that our method SAS improves the vanilla DT in terms of both reward and failure rate. It implies that our method stabilize DT and make DT show more consistent resut. In Table 8, we show our method improves the test-time adaptation performance better than APE-V which is a prior test-time adaptation method on CQL.
> > - The key contribution of our method is that SAS improves the offline safe RL method by leveraging the pretrained occupancy measure information. Staying in the expert distribution does not generalize the performance of the test-time deployment in offline RL. As shown in Table 2, CDT+SAS succesfully execute non-myopic actions better than the standard CDT by preventing from choosing the less probable actions in the long-term view. This cannot be acheived by simplying following the state-wise expert data distribution.
> >
> > > Q10: Minor issues
> > >
> >
> > Thank you for your detailed comments. We address your concern and provide the revised contents as follows.
> >
> > - Citation for Def 3.1: as mentioned in the above part of Def 3.1, the definition is from LDM paper (Kang et.al. 20222).
> > - $B^c$ is the complement set of $B$.
> > - We add the explanation of CDT in lines 106-107 in the revised version.

---

> ### Comment · Reviewer_8v92 · 2024-11-27
>
> Thank you for detailed response and explanations. I will respond to some of your points here and address the others shortly.
>
> Regarding Sec 4 and the connection to Safe RL (specifically constrained RL with a cost function, as it is presented in this section) - I continue to feel that this connection is extraneous. Specifically I am referring to Sec 4 prior to the Sec 4.1 heading. The main argument of this section is that expert data generated from a safe expert will mostly obey safety constraints and hence this will be reflected in the data distribution. However, your method is agnostic to this - your method is to stay within the high density regions of the expert distribution - the reason why the expert hasn't visited these regions is irrelevant. Moroever, the results from Sec 4 are not used anywhere in your algorithm as far as I can tell - particularly since, as you state you are not even directly controlling the $D$ parameter and hence the threshold derived in this section. I agree that combining CDT+SAS is a method for "constrained RL", however, this is not the argument made in Sec.4 which is claiming that SAS alone has connections to constrained RL. However, the only connection is through the expert demonstrations - if these are safe than SAS would be safe (with respect to the constraints), if these are not safe than SAS alone is not safe.
>
> I am saying this because I find the paper to be very busy, which makes it quite hard to follow. The paper would be significantly improved in my opinion by narrowing the focus to only the elements that contribute towards the proposed algorithm.
>
> Regarding your comment on the safety regions, while I agree that the unsafe regions could be larger than the hazards, since there may be regions that are likely to enter the hazards, this is not really what I meant when I said that the regions are not well aligned. I am referring more to the many hazards that are _not_ covered by the unsafe regions. Shouldn't the unsafe regions at a minimum cover the hazards? Why is this not the case?

---

> ### Author Response · Authors · 2024-11-28
>
> Thank you for the additional constuctive discussion. However, we respectfully disagree with your comment and provide the detailed response as follows.
>
> 1. **The connection between the safety constraint and the expert data distribution:** The key motivation to discuss this in the front of Section 4 is that we propose DT+SAS that is based on offline safe RL which uses the expert distribution and improves safety performance during test-time deployment. Without the relation of the occupancy measure of the expert distribution and the safety constraint in terms of cost, we cannot guarantee the quantitative safety performance theoretically. Equation (7) is directly connected to the cost (safety) constraint, so we can improve the safety performance of DT+SAS by guaranteeing that the agent stays in our control invariant set.
> 2. **The role of** $D$: the parameter $D$ is introduced to explain the difference of the data qaulity between the ground-truth optimal policy and the given data distribution. Since the given data distribtuion is not always the perfect expert data, we need to design the control invariant set in Equation 7 more conservatively. We first note that we can control the volume of $R_G^{SAS}$ by the definition (7) to allow the higher crietrion on the occupancy measure and the target cost. As we mentioned earlier, we have not directly used $D$, but we have already reported the experimental result related to **the conservatism by $D$ in N-trial graph** of Figure (4).  With the larger $N$ , the agent fails to search a safe trajectrion during imagination by becoming too conservative and myopic becuse of the smaller control invariant set as $E_j$ decreases. This is related to the case that the high safety constraint makes the agent converge in safe RL.
> 3. **many hazards that are *not* covered by the unsafe regions:** The unsafe regions do not cover all hazards because we illustrate the contour at the 95th percentile of $G_{SAS}$. While lowering this percentile threshold would cover all hazards, our experiments show that the agent can successfully reach the goal with low cost even in regions with a looser percentile criterion. Although all hazards are inherently dangerous, the agent demonstrates the ability to successfully navigate around hazards that lie outside the marked unsafe regions.
>
> We sincerely appreciate your detailed feedback. While our manuscript may appear dense due to the rigorous criteria required for safe RL and cost constraints, we believe this thoroughness is necessary. Please let us know if you have any additional questions or if we can clarify anything further.

---

### Meta-Review · Area_Chair_z9cg · 2024-12-19

**Metareview:**

This paper proposes a novel strategy for safe reinforcement learning which performs test-time adaptation to ensure safety guarantees (which are also provided at test time). However, there were significant concerns about the paper, including the clarity of writing (which additionally made it difficult to validate some of the theoretical claims), the positioning of the paper (in particular, whether safe reinforcement learning is the right point of comparison), and the experiments (which were on limited benchmarks). Thus, I believe the paper significant revision before it can be accepted.

**Additional Comments On Reviewer Discussion:**

The reviewers generally agreed that the paper had significant issues, especially with regards to clarity and the experiments. While the authors clarified some of these issues during the rebuttal period, the overall problems remained.

---

### Decision · Program_Chairs · 2025-01-22

Reject